# TABULA: A Tabular Self-Supervised Foundation Model for Single-Cell Transcriptomics

**Jiayuan Ding**[*†]
Stanford University

**Jianhui Lin**[*]
Central South University

**Shiyu Jiang**[*]
University of Southern California

**Yixin Wang**
Stanford University

**Ziyang Miao**
Central South University

**Zhaoyu Fang**
Central South University

**Jiliang Tang**[†]
Michigan State University

**Min Li**[†]
Central South University

**Xiaojie Qiu**[†]
Stanford University

## Abstract

Foundation models (FMs) have shown great promise in single-cell genomics, yet current approaches, such as scGPT, Geneformer, and scFoundation, rely on centralized training and language modeling objectives that overlook the tabular nature of single-cell data and raise significant privacy concerns. We present TABULA, a foundation model designed for single-cell transcriptomics, which integrates a novel tabular modeling objective and federated learning framework to enable privacy-preserving pretraining across decentralized datasets. TABULA directly models the cell-by-gene expression matrix through column-wise gene reconstruction and row-wise cell contrastive learning, capturing both gene-level relationships and cell-level heterogeneity without imposing artificial gene sequence order. Extensive experiments demonstrate the effectiveness of TABULA: despite using only half the pretraining data, TABULA achieves state-of-the-art performance across key tasks, including gene imputation, perturbation prediction, cell type annotation, and multi-omics integration. It is important to note that as public single-cell datasets continue to grow, TABULA provides a scalable and privacy-aware foundation that not only validates the feasibility of federated tabular modeling, but also establishes a generalizable framework for training future models under similar privacy-preserving settings. All resources are openly available at `https://github.com/aristoteleo/tabula` to support broad community adoption and future methodological advances.

## 1 Introduction

Over the past year, we have witnessed the emergence of single-cell foundation models, notably Geneformer [1], scFoundation [2], CellPLM [3], scGPT [4], and NicheFormer [5]. However, current models fail to explicitly account for the tabular structure of the scRNA-seq data, often naively converting gene expression within a single cell to gene sequences to mimic word sequences used in NLP. While this adapts the NLP paradigm to single-cell data, it overlooks critical features of the underlying tabular structure of scRNA-seq. Moreover, it has been shown previously that scRNA-seq data is susceptible to privacy breaches through linking attacks [6, 7] (More details are provided in Appendix D), where information from one study can be linked to another to identify private data.

---

[*] Equal contribution.  [†] Corresponding author.

39th Conference on Neural Information Processing Systems (NeurIPS 2025).

This privacy concern becomes even more pronounced as we begin creating foundation models with tens of millions of single cells and datasets from thousands of individuals.

To address these critical gaps, we introduce TABULA, a foundation model specifically designed for single-cell data through the lens of tabular learning and federated training. TABULA introduces four key innovations:

First, TABULA formulates single-cell pretraining as a self-supervised tabular learning problem, directly modeling the cell-by-gene tabular structure. It is important to note that TABULA uses corrupted gene expression inputs instead of introducing artificial mask values such as -1 in traditional FMs to learn robust gene-level representations through column-wise reconstruction, and cell-level representations through row-wise contrastive learning. This tabular modeling strategy captures both gene dependencies and cell-level heterogeneity, without imposing artificial gene order.

Second, TABULA is trained in a federated learning setting not only to preserve data privacy across institutions or tissue datasets, but also to enable tissue-specific embeddings by decoupling the training of a shared tabular transformer from client-specific embedders. This design allows TABULA to capture both globally shared biological patterns and unique transcriptional features of each client tissue.

Third, extensive experiments demonstrate the effectiveness of TABULA. TABULA achieves state-of-the-art performance across a wide range of benchmark tasks, including gene imputation, genetic perturbation prediction, cell type annotation, multi-omics integration, and batch correction while requiring only half the data for pretraining.

Fourth, as public single-cell datasets continue to grow, the need for scalable and privacy-preserving training becomes increasingly important. TABULA provides a generalizable framework that combines federated learning with tabular modeling, enabling collaborative model training without data sharing. This not only validates the feasibility of federated tabular modeling but also lays the foundation for training future models under similar privacy-preserving settings.

## 2 Design Principles

Despite recent progress in single-cell foundation models [8, 4, 1, 2], existing approaches face critical limitations that hinder their biological fidelity, scalability, and privacy. These limitations motivate the design of TABULA, which is guided by the following three principles.

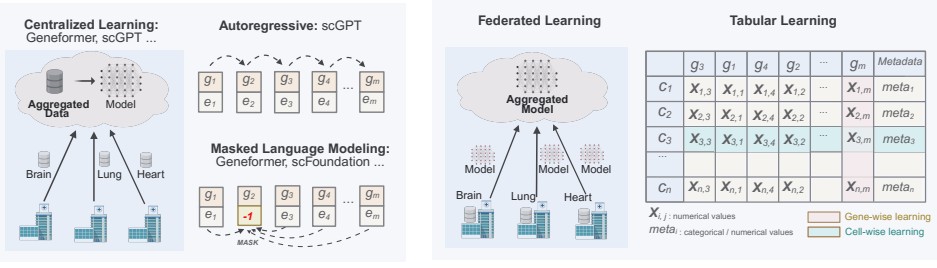

(a) Existing Foundation Models          (b) TABULA

Figure 1: An illustration of the difference between (a) existing foundation models and (b) TABULA for single-cell modeling.

**Principle 1: Realistic Corruption Strategies.** Existing FMs such as Geneformer [1] and scFoundation [2] often use artificial corruption strategies (e.g., masking gene expression values with -1) for training (Figure 1a, right). These unrealistic sentinel values do not appear during inference and result in a training-inference mismatch that degrades generalization. TABULA avoids this issue by generating corrupted views through resampling from the empirical marginal distribution of gene values (Figure 2), preserving statistical realism and improving downstream biological performance.

**Principle 2: Tabular Modeling for Unordered Mixed-Type Single-Cell Data.** Unlike natural language, where sequence order is meaningful, gene expression profiles lack an inherent ordering and

consist of mixed discrete gene token IDs and continuous gene expression values. Most existing models artificially impose gene sequences via attention masking (scGPT [4]) or ranking (Geneformer [1]) (Figure 1a, right), which distorts the tabular nature of the data. In contrast, TABULA treats the cell-by-gene matrix as a table and learns both gene-level dependencies via column-wise reconstruction and cell-level variation via row-wise contrastive learning (Figure 1b, right). This dual-axis self-supervised learning respects the true tabular structure of the data.

**Principle 3: Federated Learning for Privacy-Preserving Collaboration.** Centralized training poses major privacy risks in genomics, where individual-level data is sensitive and protected. Prior work has shown that scRNA-seq data is vulnerable to linkage attacks [6, 7]. To mitigate these risks, TABULA is trained using a federated learning setup (Figure 1b, left). Each client (e.g., tissue or institution) retains its data locally and contributes to a shared global model by only exchanging weights, enabling collaborative training across heterogeneous datasets without data sharing.

These three design principles, realistic corruption, tabular structure awareness, and federated privacy, form the foundation of TABULA and are reflected in the framework described in Section 3.

## 3 The Proposed TABULA Framework

In this section, we introduce the proposed TABULA framework and each component in TABULA. As illustrated in Figure 2, the tabular modeling in TABULA requires corrupted gene expression inputs, which are detailed in Section 3.1. The process by which the tabular embedder transforms the cell-by-gene matrix into embeddings for the transformer, along with the modeling of cell-by-gene structure from a tabular learning perspective, is detailed in Section 3.2. At a high level, TABULA consists of two stages: pre-training and fine-tuning, which are demonstrated in Section 3.3.

### 3.1 Corrupted View Generation for Tabular Modeling

To handle the intrinsic tabular structure of scRNA-seq dataset, TABULA introduces an innovative self-supervised tabular modeling framework that consists of both gene-wise reconstruction learning and cell-wise contrastive learning. Both training components start with generating a corrupted representation of the scRNA-seq data. We followed Xtab [9] to construct corrupted views through random feature resampling. As shown in Figure 2, we randomly selected a subset of genes in cells in a training batch and then resampled their values from the empirical marginal distribution [10] of these genes within the same training batch. We set the corrupted ratio at 60%. This means that for the original view of each cell sample $c$ and its corrupted view $\tilde{c}$, 60% of the gene values are resampled while 40% remain unchanged. The original and corrupted views of the same cell are treated as a positive pair in contrastive learning, as they originate from the same underlying biological state but differ through controlled noise.

### 3.2 TABULA Model Architecture

TABULA innovatively treats single-cell data, represented as a cell-by-gene matrix, as tabular data, and models its intrinsic tabular structure through a dual-axis self-supervised learning framework. Additionally, TABULA addresses cross-tissue variation by partitioning the model into distinct clients in a federated setting, each tissue client equipped with tissue-specific embedders that capture tissue-specific variants, and a shared tabular transformer (global or local) that encodes global knowledge across tissues. In this section, we describe how TABULA encodes the raw cell-by-gene matrix into transformer-compatible embeddings in **Tabular Embedder**, and how it models the matrix from a tabular learning perspective in **Tabular Modeling**.

**Tabular Embedder** scRNA-seq data can be represented as a cell-by-gene matrix, $X \in \mathbb{R}^{N \times M}$, where each entry $X_{i,j} \in \mathbb{R}^+$ denotes the read count of RNA molecules for gene $j \in \{0, 1, \ldots, M\}$ in cell $i \in \{0, 1, \ldots, N\}$. Each row of the cell-by-gene table, $X$, is considered as an input cell sample in a tabular transformer, and each column is a gene feature token. The function of the embedder module is to convert each cell sample to feature embeddings $\mathbf{E} \in \mathbb{R}^{(M+1) \times d}$. Here, $M$ denotes the number of columns, $+1$ indicates the special [CLS] token (see below), and $d$ is the embedding dimension. TABULA treats each row of the cell-by-gene matrix as an unordered set of genes, where each gene is represented by a column feature name embedding (gene token) and a feature value embedding

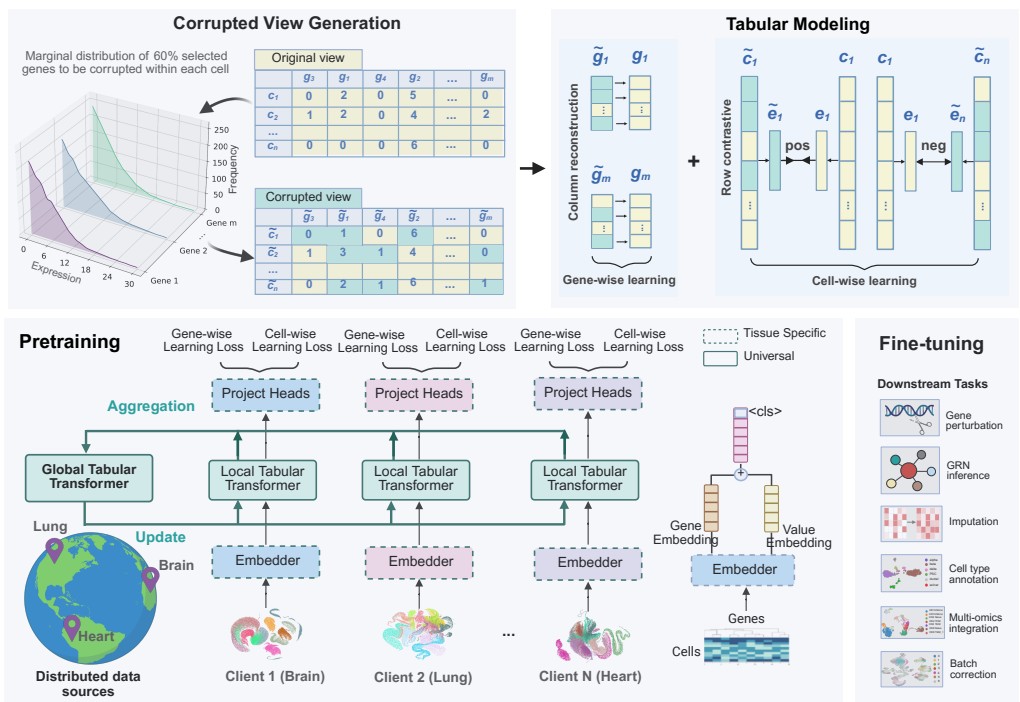

Figure 2: Overview of the TABULA framework. TABULA learns both gene- and cell-level representations from tabular single-cell data using dual-axis self-supervised tabular learning objectives: column-wise gene reconstruction and row-wise cell contrastive learning. It is trained in a federated setting with a shared global transformer and client-specific embedders, enabling privacy-preserving, and tissue-aware representation learning.

(expression level). While the current implementation models only gene tokens, the framework can naturally incorporate metadata (e.g., cell type, tissue, or platform) in future extensions.

- **Column feature name embedding.** The feature column in the cell-by-gene table for each client corresponds to individual genes. Our pretraining dataset includes 196 distinct studies, from which we select 1,200 HVGs per study, yielding a total of 23,156 HVGs across all datasets. Consequently, we have 23,156 unique feature columns to learn during the pretraining phase. We assign each column feature name, $feature_j$, a unique integer identifier, $id(feature_j)$. These identifiers collectively form the column feature name vocabulary utilized in TABULA. Additionally, a special token, [CLS], is used to aggregate all column features into a cell-level representation. The input column feature tokens for cell $i$ are therefore represented as a vector $\mathbf{f}_{\text{token}}^{(i)} \in \mathbb{R}^M$, where $M$ is a predetermined maximum sequence length in the tabular transformer:

$$\mathbf{f}_{\text{token}}^{(i)} = \left[ id\left(feature_1^{(i)}\right), id\left(feature_2^{(i)}\right), \ldots, id\left(feature_M^{(i)}\right) \right] \quad (1)$$

The embedding representation of input column feature tokens for cell $i$ is denoted as $\mathbf{E}_{\text{token}}^{(i)} \in \mathbb{R}^{M \times d}$ where $d$ represents the embedding dimension:

$$\mathbf{E}_{\text{token}}^{(i)} = \text{emb}_{\text{token}}\left(\mathbf{f}_{\text{token}}^{(i)}\right) \quad (2)$$

where $\text{emb}_{\text{token}}$ represents an embedding layer to learn feature token embeddings.

- **Column feature value embedding.** Expression values in scRNA-seq data vary due to differences in sequencing protocols, sensitivity, and depth, leading to inconsistencies across datasets. To tackle this issue, we adopt scGPT's value binning technique [4] to convert expression counts into relative values. Unlike scGPT, which only focuses on non-zero expression counts in each cell, we select 1,200 HVGs for modeling in each dataset, including those with zero expression.

By incorporating zero-expressed genes, TABULA provides a more complete view of gene expression variability, capturing both active and inactive genes. This broader perspective allows for the detection of important biological patterns that could be overlooked when considering only non-zero genes, as zero expression can have biological significance [11]. For each HVG expression count in a cell, we compute the raw absolute values and partition them into $B$ consecutive intervals $[Bin_t, Bin_{t+1})$, where $t \in \{1, 2, \ldots, B\}$. The binned expression value $x_j^{(i)}$ for cell $i$ is designated as:

$$x_j^{(i)} = \begin{cases} t, & \text{if } X_{i,j} > 0 \text{ and } X_{i,j} \in [Bin_t, Bin_{t+1}) \\ 0, & \text{if } X_{i,j} = 0 \end{cases} \tag{3}$$

The column feature values for cell $i$ are represented as a vector $\mathbf{f}_{\text{val}}^{(i)} = \left[ x_1^{(i)}, x_2^{(i)}, \ldots, x_M^{(i)} \right] \in \mathbb{R}^M$, where $M$ is the predefined maximum sequence length. These values are then embedded via $\mathbf{E}_{\text{val}}^{(i)} = \text{emb}_{\text{val}}(\mathbf{f}_{\text{val}}^{(i)})$, where $\mathbf{E}_{\text{val}}^{(i)} \in \mathbb{R}^{M \times d}$ and $\text{emb}_{\text{val}}$ is a learnable embedding layer.

Consequently, the embedding $\mathbf{E}^{(i)} \in \mathbb{R}^{M \times d}$ for cell $i$ is represented as:

$$\mathbf{E}^{(i)} = \mathbf{E}_{\text{token}}^{(i)} + \mathbf{E}_{\text{val}}^{(i)} \tag{4}$$

Finally, the embedding of the [CLS] token is appended to the feature embedding, resulting in a final feature embedding $\mathbf{E}^{(i)} \in \mathbb{R}^{(M+1) \times d}$. The final hidden state corresponding to the [CLS] token is used as row (cell) representation, aggregating from column feature embeddings. It is important to note that each client's embedder module is distinct and specialized for learning tissue-specific features.

**Tabular Modeling.** To effectively model the cell-by-gene table, we design two specialized pretraining losses for tabular modeling: reconstruction loss and contrastive learning to learn gene column features and row cell features, respectively shown in Figure 2. To calculate the reconstruction and contrastive losses, a corrupted view must be generated for each cell, which has been discussed in Section 3.1. To note that the original view $c$ and corrupted view $\tilde{c}$ of the same cell are treated as a positive pair in contrastive learning, as they originate from the same underlying biological state but differ through controlled noise.

- **Reconstruction loss as gene-wise column learning.** Reconstruction loss is a self-supervised training objective. It is important to note that it aims to recover the original gene-wise view from a corrupted view of that gene across all cells in the batch. Taking gene $j$ as an example, we take the corrupted view, $\tilde{\mathbf{c}}_j \in \mathbb{R}^N$ ($N$ indicates the number of cells in a batch.), for gene $j$ as input and aim to reconstruct its original binned gene expression $\mathbf{g}_j \in \mathbb{R}^N$ where $\mathbf{g}_j = \left[ c_j^{(1)}, c_j^{(2)}, \ldots, c_j^{(N)} \right]$. The reconstructed view for gene $j$ can be represented as $\hat{\mathbf{g}}_j \in \mathbb{R}^N$. In this study, we use Mean Squared Error (MSE) to measure the distance between the original view and the reconstructed view. The reconstruction loss is denoted as:

$$\mathcal{L}_{\text{rec}} = \frac{1}{M \cdot N} \sum_{j=1}^{M} \sum_{i=1}^{N} \left( \hat{c}_j^{(i)} - c_j^{(i)} \right)^2 \tag{5}$$

The reconstruction loss $\mathcal{L}_{\text{rec}}$ is applied to all $M$ gene values, enabling the model to learn gene-specific distributions and inter-gene relationships for richer feature representations. Unlike prior models [8, 2, 1, 4] that use an artificial mask value (e.g., -1), we replace corrupted entries by resampling from the empirical gene distribution, aligning training conditions with inference for improved consistency.

- **Contrastive loss as cell-wise row learning.** Similar to the reconstruction objective, we generate $\tilde{c}^{(i)}$ as a corrupted view of cell $c^{(i)}$. The original and corrupted views of the same cell form a positive pair, while all other cells ($\mathbf{c}^{(j)}$ and $\tilde{\mathbf{c}}^{(j)}$) within the batch are treated as negative pairs. Contrastive loss encourages the model to minimize the distance between positive pairs and maximize it between negatives. Specifically, as shown in Figure 2, the gene expression profiles of each cell, including the original view $c_1$ and corrupted view $\tilde{c}_1$, are embedded into latent representations ($\mathbf{e}_1, \tilde{\mathbf{e}}_1$). The loss enforces that positive pairs (e.g., $\mathbf{e}_1$ and $\tilde{\mathbf{e}}_1$) are close in latent space, while negative pairs (e.g., $\mathbf{e}_1$ and $\tilde{\mathbf{e}}_n$) are far apart.

In this study, we employ the SimCLR [12] loss for contrastive pretraining, which is denoted as:

$$\mathcal{L}_{\text{contrast}} = -\frac{1}{2N} \sum_{i=1}^{N} \left[ \log \frac{\exp\left(\text{sim}(\mathbf{e}_i, \tilde{\mathbf{e}}_i)/\tau\right)}{\sum_{j=1}^{N} \mathbb{I}_{[j \neq i]} \left(\exp\left(\text{sim}(\mathbf{e}_i, \tilde{\mathbf{e}}_j)/\tau\right) + \exp\left(\text{sim}(\mathbf{e}_i, \mathbf{e}_j)/\tau\right)\right)} \right.$$
$$\left. + \log \frac{\exp\left(\text{sim}(\tilde{\mathbf{e}}_i, \mathbf{e}_i)/\tau\right)}{\sum_{j=1}^{N} \mathbb{I}_{[j \neq i]} \left(\exp\left(\text{sim}(\tilde{\mathbf{e}}_i, \mathbf{e}_j)/\tau\right) + \exp\left(\text{sim}(\tilde{\mathbf{e}}_i, \tilde{\mathbf{e}}_j)/\tau\right)\right)} \right] \quad (6)$$

where $\mathbf{e}^{(i)}$ and $\tilde{\mathbf{e}}^{(i)}$ represent the cell embedding of cell $i$ from the original view and corrupted view respectively after being processed through the transformer block. $\mathbb{I}_{[k \neq i]} \in \{0, 1\}$ is an indicator function that takes the value of 1 if $k \neq i$; otherwise, it is 0. The similarity function $\text{sim}(\cdot, \cdot)$ represents cosine similarity here to measure the similarity between two cell embeddings, $\tau$ denotes the temperature parameter, and $N$ is the number of cells in a batch.

The final tabular modeling objective in TABULA is a combination of the two loss functions:

$$\mathcal{L}_{\text{tab}} = \alpha \mathcal{L}_{\text{rec}} + \mathcal{L}_{\text{contrast}} \quad (7)$$

where the scaling factor $\alpha$ is used to balance the two loss terms to ensure comparable magnitudes. In this study, it is set to 0.03. The combined loss provides a holistic approach to learning both gene-level (column-wise) and cell-level (row-wise) features from the cell-by-gene matrix. While gene-wise reconstruction promotes robust gene-level representation learning, cell-wise contrastive learning enforces both intra-cell consistency and inter-cell variability. Together, these objectives enable TABULA to effectively model the tabular structure of single-cell data across both the gene and cell axes.

### 3.3    TABULA Pre-training & Fine-tuning

**Pre-training.** To protect data privacy in single-cell data, TABULA is pretrained using a federated learning framework shown in Figure 2. Each client, representing a hospital, research institute, or tissue type (e.g., lung, brain, heart), trains locally without sharing raw data. In this study, TABULA is pretrained on 15 million scRNA-seq profiles across eight clients: Intestine, Pancreas, Lung, Heart, Blood, Kidney, Brain, and Others (see Appendix A for pretraining data collection and preprocessing). Each client includes a tissue-specific embedder that encodes the cell-by-gene matrix into embeddings. These embeddings are passed through a tabular transformer to compute gene-wise and cell-wise self-supervised losses (shown in Equation 7). Training proceeds in alternating rounds of local and global updates: local transformer weights from client-side are uploaded to a central server, aggregated into a global model, and then broadcast back to clients (see Appendix B for the federated learning framework). This iterative communication enables knowledge sharing across heterogeneous datasets while preserving data confidentiality through strict decentralization. Note that this cross-tissue client setup demonstrates federated training of TABULA, the framework is generally applicable to cross-institutional or other settings, enabling collaborative pretraining without sharing raw data and thereby addressing privacy concerns. More pertaining details can be found in Appendix C.

**Task-specific Fine-tuning.** TABULA can be fine-tuned for a variety of downstream tasks, including gene perturbation prediction, cell type annotation, multi-omics integration, batch correction, and gene expression imputation shown in Figure 2. During fine-tuning, it first loads the tissue-specific pretrained embedder, followed by the pretrained global tabular transformer. A task-specific head or decoder is then attached based on the nature of the task, enabling the model to adapt effectively through task-specific optimization.

# 4 Experiments

## 4.1 Preliminary Study: Federated vs. Centralized and Tabular vs. MLM

To demonstrate the advantage of TABULA 's tabular learning and its adaptability to federated learning, we compare four pretraining strategies: centralized vs. federated learning, each paired with either tabular modeling or masked language modeling (MLM) (Figure 3 (left)). The pretraining dataset includes 1M cells from CELLxGENE (250K per tissue: pancreas, blood, brain, lung). Centralized training uses pooled data; federated training distributes data across four tissue clients with model weights aggregation. MLM uses a 15% masking ratio. We assess performance on two downstream tasks: cell type annotation (hPancreas dataset [13]) and genetic perturbation prediction (Adamson [14] and Norman [15] datasets), measuring accuracy and Pearson correlation coefficients. As shown in Figure 3, the results show that tabular modeling consistently outperforms MLM in both training schemes. Notably, federated-tabular pretraining achieved the highest overall performance, matching or exceeding centralized models despite data heterogeneity. More details of the implementation can be found in the Appendix C.

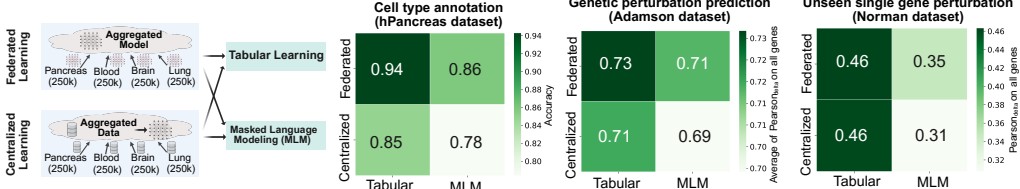

Figure 3: Benchmarking federated vs. centralized pretraining and tabular learning vs. masked language modeling (MLM) on 1 million cells.

## 4.2 Pretraining Data Scaling Laws in TABULA

To investigate how pretraining dataset size influences downstream performance, we identify a clear scaling law exhibited by TABULA. As depicted in Figure 4, increasing the number of pretraining cells from 300,000 to 3 million and then 15 million progressively enhances cell type annotation accuracy in the hPancreas dataset [13]. UMAP visualizations (Figure 4a) reveal more coherent clustering of annotated cell populations as data scale increases, especially notable in the alpha and beta cell clusters (highlighted in red). Quantitative analysis (Figure 4b) further confirms consistent improvements across accuracy, precision, recall, and macro F1 score, highlighting that larger-scale pretraining substantially strengthens the generalizability and robustness of learned representations. Importantly, these results also demonstrate that the benefits of scaling laws hold even under a federated learning framework, underscoring TABULA 's ability to leverage large-scale, decentralized datasets while preserving data privacy.

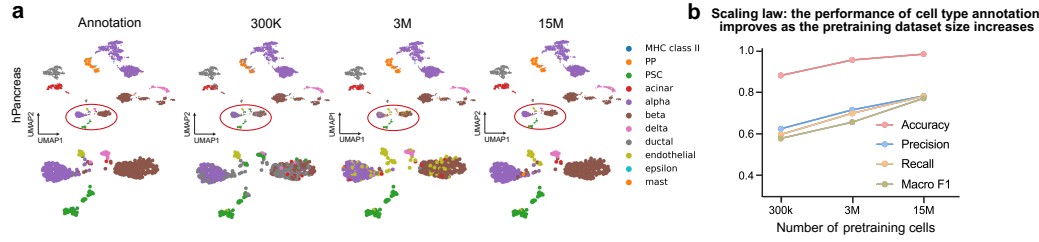

Figure 4: **Scaling law in TABULA. (a)** Pancreas-specific cell type annotation performance on the hPancreas dataset [13], visualized via UMAP. **(b)** Scaling law showing improved cell type annotation performance (accuracy, precision, recall, and macro F1) as the pretraining dataset increases from 300K to 3M and 15M cells.

## 4.3 Evaluation on Gene-Level Downstream Tasks

**Task 1: Gene Imputation** We first summarize key differences among TABULA and state-of-the-art foundation models from training paradigm, modeling objective, and pretraining data scale as shown

in Figure 5a. Unlike existing centralized models that rely on language modeling objectives (AR or MLM), TABULA adopts a federated training mechanism and explicitly models the tabular structure of single-cell data with only half the pertaining data. We evaluate gene-level imputation performance by masking non-zero expression values in scRNA-seq data and predicting them using a fine-tuned decoder. TABULA is benchmarked against scGPT, scBERT, and Geneformer across four datasets (PBMC5K, Jurkat, Melanoma [16], hPancreas [13]). As shown in Figure 5b, TABULA consistently achieves lower mean squared error (MSE) at both the cell and gene levels, with statistically significant improvements (Wilcoxon $P < 0.05$, marked by *). See Appendix E for additional results and implementation details.

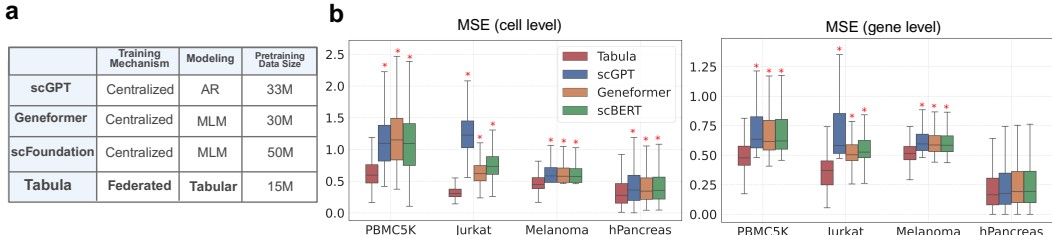

Figure 5: TABULA improves imputation fidelity and accuracy. (a) Differences between TABULA and state-of-the-art foundation models. (b) MSE comparisons across four datasets at cell and gene levels; TABULA consistently achieves lower errors (Wilcoxon $P < 0.05$, marked by *).

**Task 2: Genetic Perturbation Prediction** We evaluate genetic perturbation prediction on three benchmark datasets: Adamson [14], Norman [15], and Replogle [17], covering both single- and double-gene perturbations. Performance is measured using Pearson$_{delta}$ between predicted and observed gene expression changes, on all genes and the top 20 differentially expressed (DE) genes per condition. We compare TABULA against both task-specific baselines, GEARS [18], CPA [19], and linear regression (LR), as well as foundation models including scGPT [4], Geneformer [1], and scFoundation [2]. As shown in Table 1, on Adamson, TABULA achieves the highest score on all genes (0.695) and performs comparably on DE genes (0.787 vs. 0.789 for scGPT). On Norman, it achieves the best DE score (0.778) and remains competitive overall. On Replogle, TABULA outperforms scGPT greatly by +0.069 (all genes) and +0.012 (DE genes). To assess generalization, we test on unseen Norman perturbations. TABULA outperforms scGPT in Pearson$_{delta}$ for both single- and double-gene settings, especially on DE genes (Figure 6 left).

We further assess reverse perturbation prediction following the setup of scGPT [4], where the task is to recover the causal gene perturbation(s) that led to an observed expression profile. We evaluate on a 20-gene benchmark comprising 210 possible single- and double-gene combinations, reporting top-K hit rates. As shown in Figure 6 (middle), TABULA achieves a 2/2 hit rate of 48.57% and a 1/2 hit rate of 94.29% at top-1, substantially outperforming scGPT. The heatmap in Figure 6 (right) visualizes the perturbation pairs correctly identified in the top-1 (red) and top-3 (blue), demonstrating TABULA 's effectiveness in pinpointing causal gene combinations. These results highlight TABULA 's strong performance in both predictive and reverse perturbation tasks, especially under distribution shifts and combinatorial settings. See Appendix E for additional results and implementation details.

Table 1: *(Task 2)* The results of genetic perturbation prediction across Adamson, Norman, and Replogle datasets, evaluated by Pearson$_{delta}$ on all genes and differentially expressed (DE) genes. OOT indicates out-of-time.

| Model | Adamson | | Norman | | Replogle | |
|---|---|---|---|---|---|---|
| | All Genes | DE Genes | All Genes | DE Genes | All Genes | DE Genes |
| GEARS | 0.531 | 0.678 | 0.547 | 0.715 | 0.154 | 0.277 |
| CPA | 0.145 | 0.309 | – | – | – | – |
| LR | 0.387 | 0.620 | 0.532 | 0.697 | 0.141 | 0.298 |
| Geneformer | 0.199 | 0.666 | 0.316 | 0.326 | 0.091 | 0.362 |
| scFoundation | OOT | OOT | OOT | OOT | OOT | OOT |
| scGPT | 0.615 | **0.789** | **0.583** | 0.742 | 0.242 | 0.464 |
| TABULA | **0.695** | 0.787 | 0.577 | **0.778** | **0.311** | **0.476** |

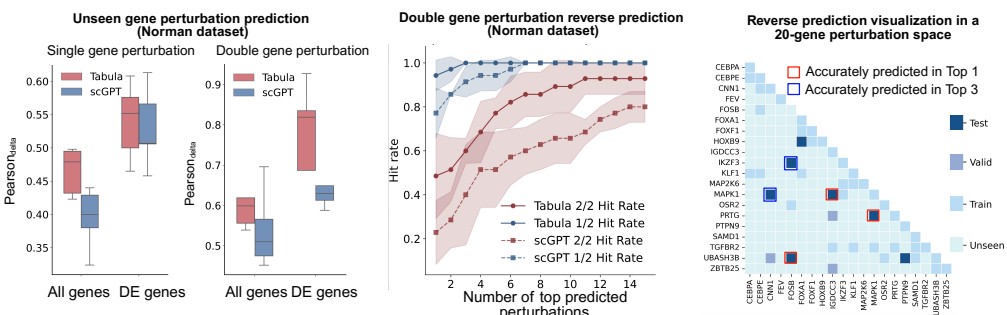

Figure 6: TABULA outperforms scGPT on unseen gene perturbation and reverse perturbation. Left: Pearson correlations for unseen single and double perturbation prediction on all and DE genes. Middle: Top-K hit rates for reverse prediction of double perturbations (1/2 and 2/2 correct). Right: Heatmap of 20-gene pairwise space with top-1 (red) and top-3 (blue) accurate reverse predictions.

## 4.4 Evaluation on Cell-Level Downstream Tasks

**Task 3: Cell Type Annotation** Cell type annotation assigns biological labels to cells based on gene expression profiles. To evaluate TABULA, we fine-tune it with a supervised cross-entropy loss and test on three datasets: hPancreas [13], Myeloid [4], and Cell Lines [20]. We compare against scGPT, Geneformer, and scFoundation, each fine-tuned with their default settings. Performance is measured using accuracy, precision, recall, and F1 score. TABULA achieves the best overall performance on hPancreas and Cell Lines, and while slightly behind scFoundation in Myeloid accuracy, it outperforms others in remaining metrics—demonstrating strong generalization (Table 2). Notably, TABULA achieves this using only half the pretraining data while ensuring privacy protection through federated training. See Appendix E for additional results and implementation details. We also report **Task 4&5 Multi-Omics and Multi-Batch Integration** results with implementation details in Appendix E.

Table 2: *(Task 3)* The results of cell type annotation across hPancreas, Myeloid, and Cell Lines datasets, evaluation by test accuracy, precision, recall, and F1 score.

| Model | hPancreas | | | | Myeloid | | | | Cell Lines | | | |
|---|---|---|---|---|---|---|---|---|---|---|---|---|
| | Accuracy | Precision | Recall | F1 | Accuracy | Precision | Recall | F1 | Accuracy | Precision | Recall | F1 |
| geneFormer | 0.9395 | 0.6534 | 0.6491 | 0.6425 | 0.5953 | 0.3704 | 0.3300 | 0.3441 | 0.9911 | 0.9918 | 0.9904 | 0.9910 |
| scFoundation | 0.9492 | 0.6140 | 0.6032 | 0.6003 | 0.6352 | **0.4070** | **0.3764** | **0.3804** | 0.9931 | 0.9932 | 0.9930 | 0.9931 |
| scGPT | 0.9680 | 0.7350 | 0.7250 | 0.7180 | **0.6420** | 0.3660 | 0.3470 | 0.3460 | 0.9930 | 0.9930 | 0.9920 | 0.9930 |
| TABULA | **0.9810** | **0.7814** | **0.7795** | **0.7708** | 0.6213 | 0.3909 | 0.3634 | 0.3728 | **0.9935** | **0.9939** | **0.9931** | **0.9935** |

In addition to these main results, we provide additional analyses in the appendix to further demonstrate the efficacy of TABULA. Specifically, we demonstrate that tissue-specific embedders can effectively capture biologically meaningful variation across different tissue types, as detailed in Appendix F. Furthermore, ablation studies presented in Appendix G confirm the importance and effectiveness of our proposed tabular modeling objectives, highlighting their critical role in enhancing downstream performance.

## 5 Related Work

Recent advances in single-cell foundation models (FMs) have adapted techniques from natural language processing to learn transferable representations from large-scale transcriptomic data. Notable models include Geneformer [1], scFoundation [2], scGPT [4], and CellPLM [3]. These models predominantly employ centralized training and language modeling objectives such as autoregressive (AR) [21] or masked language modeling (MLM) [22], which ignore the intrinsic tabular structure of single-cell data. Moreover, those models reply on centralized training thus raising concerns about data privacy and ethics from public single-cell data. Prior work has shown that scRNA-seq data is vulnerable to linkage attacks [6, 7].

To address these limitations, we propose TABULA, a foundation model that captures the tabular structure of single-cell data through tabular modeling while preserving privacy through federated learning. TABULA employs a dual-axis pretraining objective, gene-wise reconstruction and cell-wise

contrastive learning, to model the cell-by-gene matrix without imposing artificial gene order. By decoupling global updates from client-specific embedders, it supports privacy-preserving learning across institutions. Extensive experiments demonstrate the effectiveness of TABULA.

## 6  Conclusion

In this work, we present TABULA, the first privacy-preserving foundation model for single-cell transcriptomics that explicitly models the cell-by-gene matrix using a novel tabular pretraining objective. To our knowledge, TABULA is the first to propose a tabular modeling framework tailored for single-cell data, moving beyond sequence-based approaches that impose artificial gene sequence order. It achieves strong performance across diverse benchmark tasks using only half the pretraining data and captures complex biological signals across gene- and cell-level representations. It is important to note that as public single-cell datasets continue to grow, TABULA offers a scalable and privacy-aware foundation that not only validates the feasibility of federated tabular modeling, but also establishes a generalizable framework for training future models under similar privacy-preserving settings.

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
