# OpenReview forum: "Tabula: A Tabular Self-Supervised Foundation Model for Single-Cell Transcriptomics"
_NeurIPS.cc/2025/Conference — NeurIPS 2025 poster_

### Official Review · Reviewer_Xu6M · 2025-06-26

**Clarity:** 3
**Significance:** 3
**Originality:** 2
**Rating:** 4
**Confidence:** 3

**Summary:**

The authors address two important challenges in current single-cell foundation models: (1) the neglect of the inherent tabular nature of transcriptomic data, and (2) serious privacy concerns related to centralized model training. To this end, they propose TABULA, a large-scale foundation model for single-cell transcriptomics that integrates table-aware modeling with federated learning. TABULA directly models the cell-by-gene expression matrix, capturing both inter-gene and inter-cell relationships. Overall, the work avoids simply imposing NLP paradigms (e.g., token ranking) on transcriptomics and instead leverages the structural characteristics of the data. The incorporation of federated learning is timely and relevant for safeguarding sensitive biomedical datasets.

**Questions:**

1. The paper claims that the model captures gene–gene relationships, but does not provide downstream analysis to support this. Prior work, such as Geneformer, evaluates the learned embeddings by analyzing similarities between transcription factor targets (both direct and indirect) and housekeeping genes. Similar evaluations could help validate gene-level representation learning [1].

2. The use of raw count inputs raises concerns about batch effects caused by tissue heterogeneity and differences in sequencing platforms. How were these issues mitigated during pretraining? Was any normalization or correction applied?

3. How does TABULA defend against backdoor or poisoning attacks in the federated setting? Has any robustness analysis or adversarial simulation (e.g., following [2]) been conducted?

4. The scaling factor is fixed at 0.03 throughout the paper. Was this value tuned empirically? How does model performance vary with different scaling factors?

5. How does TABULA differ from existing tabular foundation models such as TabPFN[3] in terms of architecture and design? Was any empirical comparison conducted to validate the design choice over directly adapting those architectures?

6. During pretraining and fine-tuning, the maximum input sequence length M varies (e.g., 1200 vs. 2000). Was this choice guided by any empirical observation? Is there any recommendation for selecting M based on dataset characteristics?

[1] Cen et al., “Transfer learning enables predictions in network biology.” Nature, 2023.

[2] Feng et al., “Unveiling potential threats: backdoor attacks in single-cell pre‑trained models.” Cell Discovery, 2024. doi:10.1038/s41421-024-00753-1

[3] Hollmann et al., “Accurate predictions on small data with a tabular foundation model.” Nature, 2025 (vol. 637, pp.319–326). doi:10.1038/s41586-024-08328-6

**Ethical Concerns:**

["NO or VERY MINOR ethics concerns only"]

**Final Justification:**

Thank you to the authors for the detailed and thoughtful response. The clarifications provided regarding the downstream task on gene-gene relationships convincingly addressed my initial concerns, significantly improving the clarity and strengthening the validity of the results. As a result of the comprehensive responses and the corresponding manuscript revisions, I find the contribution of the paper to be more substantial and well-supported. Therefore, I am updating my score to 4.

**Limitations:**

yes

**Quality:**

2

**Strengths And Weaknesses:**

Strengths

1. The paper is generally well-written and well-organized. The authors provide comprehensive benchmarks covering both gene-level and cell-level downstream tasks, demonstrating the versatility and broad applicability of the TABULA.
2. The model appropriately respects the tabular format of single-cell data and employs a Transformer to capture inter-gene relationships.
3. Despite being trained on only half of the available training data, the model appears to achieve good performance across multiple tasks.

Weaknesses
1. In the federated pretraining setup, model weights are aggregated via FedAvg using client-wise weights $p_k$. The paper states that equal weighting is used during pretraining. However, in downstream applications where users contribute their own models for weight updates, it is unclear how $p_k$ is determined or whether adaptive weighting is supported. This should be explicitly clarified.
2. The current model architecture does not incorporate attention across cells—attention is computed only over genes. This design limits the model's ability to capture inter-cellular relationships, which are critical in tasks such as trajectory inference.
3. The strategy for selecting highly variable genes (HVGs) in the federated learning setting is under-specified. Are HVGs selected independently per client? Was any study conducted to understand how the number of HVGs (M) affects model performance?
4. Although the paper highlights the use of federated learning to enhance privacy, it does not address potential security vulnerabilities, such as backdoors or data poisoning attacks, which are well-documented threats in federated systems.

---

> ### Author Rebuttal · Authors · 2025-07-31
>
> **Question 1:** The paper claims to capture gene–gene relationships but lacks downstream analysis to support this.
>
> "Answer:"  To demonstrate that Tabula captures gene–gene relationships, we evaluate its zero-shot performance on two tasks: (1) Pairwise regulation – identifying regulatory effects between two genes; and (2) Combinatorial regulation – assessing joint effects of multiple genes on a target.
>
> For pairwise validation, we evaluated four developmental systems—hematopoiesis (29 known pairs), cardiogenesis (17), neurogenesis (9), and pancreatic endogenesis (22)—and compared predictions to known regulatory relationships. Results are presented in Table 1.
>
> **Table 1:** Pairwise regulation prediction accuracy from Tabula across four developmental systems.
>
> | System                 | Accuracy (%) |
> |------------------------|--------------|
> | Hematopoiesis          | 0.83         |
> | Cardiogenesis          | 0.88         |
> | Neurogenesis           | 0.89         |
> | Pancreatic Endogenesis | 0.91         |
>
> We tested combinatorial regulation prediction on hematopoiesis and cardiogenesis, which contain 4 and 9 known combinatorial relationships, respectively. Taking CR1(8) from the cardiogenesis system as an example, its regulatory rule is defined as: FGF8 = MESP1 ∩ (FOXC1 ∪ TBX1), indicating a three-gene combinatorial regulation with 8 possible regulatory combinations. The accuracy results are shown in Table 2.
>
> Across both pairwise and combinatorial regulation tasks, Tabula demonstrates strong **zero-shot performance** spanning four developmental biological systems. These results highlight Tabula’s ability to capture gene–gene relationships during pretraining.
>
> **Table 2:** Combinatorial regulation prediction accuracy from Tabula across two developmental systems.
>
> | System        | Combinatorial Relationship | Accuracy (%) |
> |---------------|---------------|---------------|
> | Cardiogenesis | CR1 (8)       | 100           |
> |               | CR2 (8)       | 100           |
> |               | CR3 (8)       | 87.5          |
> |               | CR4 (8)       | 12.5          |
> | Hematopoiesis | CR1 (16)      | 100           |
> |               | CR2 (16)      | 100           |
> |               | CR3 (16)      | 81.25         |
> |               | CR4 (16)      | 75            |
> |               | CR5 (8)       | 87.5          |
> |               | CR6–CR9 (4)   | 100 (each)    |
>
>
> **Question 2:** The use of raw count inputs raises concerns about batch effects caused by tissue heterogeneity and differences in sequencing platforms. How were these issues mitigated during pretraining? Was any normalization or correction applied?
>
> **Answer:** This is an excellent question. This is a fundamental challenge for single-cell foundation models.
>
> We clarify that Tabula does not use raw count data directly. Instead, following scGPT’s binning strategy (see **Column feature value embedding**), we discretize expression values into 50 bins, computed independently per cell to reduce batch effects and platform variability.  Additionally, rather than scGPT’s random selection of 1,200 non-zero genes per cell, we use 1,200 dataset-specific HVGs, enabling more targeted and efficient learning.
>
> **Question 3:** How does TABULA defend against backdoor or poisoning attacks in the federated setting? Has any robustness analysis or adversarial simulation (e.g., following [1]) been conducted?
>
> **Answer:** Thank you for this important question on Tabula’s robustness in federated settings.
>
> **1. Backdoor Defense in Pretraining**: Tabula uses self-supervised learning on unlabeled data, which inherently reduces vulnerability to traditional backdoor attacks, as adversaries cannot manipulate labels to inject specific triggers.
>
> **2. Robustness Analysis**: Following [1], we simulated label poisoning attacks during downstream fine-tuning (note: no federated learning is used here) by introducing abnormal cell type labels. To detect anomalies, we extended the get_abnormal_cells method from [1], adding K-means clustering on abnormality scores for automatic identification. Evaluated on the human pancreas dataset [2], our method achieved 96.53 ± 0.96% recall and 99.38 ± 1.47% precision, demonstrating strong first-pass defense. We will clarify these robustness efforts in the revision.
>
> We are committed to proactively integrating privacy and security safeguards as the ecosystem evolves and plan to conduct systematic adversarial simulations in future work to comprehensively assess Tabula’s robustness against emerging attack vectors.
>
> **Question 4:** The scaling factor is fixed at 0.03 throughout the paper. Was this value tuned empirically? How does model performance vary with different scaling factors?
>
> **Answer**: We really appreciate your thoughtful question. We fix α=0.03 so that reconstruction and contrastive losses have similar magnitudes - this stabilizes optimization. Please see our response to **Reviewer bEnN’s Question 1** for more details.
>
> **Question 5:** How does TABULA differ from existing tabular foundation models such as TabPFN[3] in terms of architecture and design? Was any empirical comparison conducted to validate the design choice over directly adapting those architectures?
>
> **Answer:** Tabula fundamentally differs from TabPFN in pretraining objective, learning setting, and applicability to biological data.
>
> **Training Objective**:
> Tabula uses a dual self-supervised approach—column-wise reconstruction and row-wise contrastive learning—tailored for sparse, high-dimensional single-cell data. In contrast, TabPFN relies on supervised in-context learning with synthetic tasks, assuming fixed feature order and semantics, which do not hold in single-cell settings.
>
> **Federated Training**:
> Tabula supports federated learning while TabPFN lacks any federated or privacy-preserving mechanisms.
>
> Due to these limitations, TabPFN is not adaptable for single-cell data, while Tabula's design enables biologically meaningful representation learning across both gene and cell axes.
>
> **Question 6:** During pretraining and fine-tuning, the maximum input sequence length M varies (e.g., 1200 vs. 2000). Was this choice guided by any empirical observation? Is there any recommendation for selecting M based on dataset characteristics?
>
> **Answer:** We thank the reviewer for the thoughtful question. We kindly refer the reviewer to our response to Weakness 3 for further clarification.
>
> **Weakness 1**: The paper uses equal weighting in FedAvg during pretraining, but it's unclear how model updates are handled in downstream tasks. Is adaptive weighting supported?
>
> **Answer:** Thank you for the thoughtful question. During federated pretraining, we use FedAvg with equal client-wise weighting (see **Appendix B**). In downstream tasks, federated learning is not applied—users fine-tune the pretrained tissue embedder and transformer locally using standard supervised training without any aggregation (see **Appendix E**). We will clarify this distinction in the revised manuscript.
>
> **Weakness 2**: The model computes attention only over genes, not across cells, limiting its ability to capture inter-cellular relationships important for tasks like trajectory inference.
>
> **Answer:** Thank you for the insightful comment. Like most foundation models (e.g., Geneformer, scGPT), Tabula treats genes as tokens to learn robust gene-level embeddings. While effective, we agree that modeling cell-cell relationships is important. Tabula 2.0 will incorporate spatial coordinates and potentially a spatial loss. For multi-omics pretraining, see our response to Reviewer bEnN’s Question 4. We will clarify this in the revision.
>
> **Weakness 3**:  Is HVG selection done independently per client in the federated setting? Was the impact of the number of HVGs (M) on model performance studied?
>
> **Answer:**  Thank you for raising this important question.
>
> **1. HVG Selection Strategy**: In our framework, each client handles multiple tissue-specific datasets, and we perform independent HVG selection per dataset, choosing the top 1,200 HVGs locally (see Appendix A). Unlike methods like scGPT, which randomly sample from non-zero genes, Tabula selects dataset-specific HVGs to ensure that biologically informative genes are used. This strategy improves training efficiency and preserves the unique biological signals of each dataset.
>
> **2. Choice of HVG Count (M)**: We chose 1,200 HVGs based on precedent in single-cell foundation models: scGPT uses 1,200 randomly selected genes [3], and Geneformer uses 2,048 based on expression ranking [4]. Standard scRNA-seq workflows typically use 1k–5k HVGs [5]. While larger M values could enhance model capacity, they introduce significant computational overhead and would require redesigning parts of the architecture. Our choice reflects a practical balance between biological richness and efficiency.
>
> **Weakness 4**: The paper emphasizes privacy via federated learning but does not address security risks like backdoors or data poisoning—how are these threats handled?
>
> **Answer:** Thank you for highlighting this important point. As the first work on federated pretraining for single-cell foundation models, our focus was on demonstrating feasibility. We agree that backdoor and poisoning attacks are important. The structured nature of single-cell data may pose unique vulnerabilities but also enable biologically informed defenses. We will note this in the revision and leave it for future work.
>
> [1] Feng et al., “Unveiling potential threats: backdoor attacks in single-cell pre‑trained models.” Cell Discovery, 2024.
>
> [2] Chen, Jiawei, et al. TOSICA, Nature Communications.
>
> [3]Cui, Haotian, et al. scGPT, Nature Methods
>
> [4]Theodoris, Christina V., et al. Geneformer, Nature
>
> [5]Luecken, Malte D., and Fabian J. Theis. "Current best practices in single‐cell RNA‐seq analysis: a tutorial." Molecular systems biology

---

> > ### Author Response · Authors · 2025-08-05
> > **Friendly Follow-Up on Tabula – Awaiting Your Feedback**
> >
> > Dear Reviewer,
> >
> > Thank you for the encouraging feedback.  We’re pleased that you value Tabula’s tabular-aware pretraining objective design; even with only half the training data, Tabula matches or surpasses state-of-the-art foundation models on diverse cell- and gene-level tasks.
> >
> > **We believe our point-by-point responses clearly resolve the concerns you highlighted in your questions and weaknesses. For additional details, please see our full replies in the last response.**
> >
> > Due to limited space in our previous response, we only presented the results of the Pairwise and Combinatorial Regulation prediction tasks used to validate Tabula’s ability to capture gene–gene relationships, without detailing how Tabula performs these tasks. Below, we provide more details:
> >
> > ---
> >  **Task 1: Pairwise Regulation Prediction**
> >
> > To infer how a **regulator gene** \( g_1 \) affects a **target gene** \( g_2 \), Tabula performs iterative *in silico* perturbation of \( g_1 \) and observes downstream changes in \( g_2 \)'s expression across cells.
> >
> > 1. **Iterative Perturbation of \( g_1 \):**
> >    We simulate a perturbation by modifying the expression of \( g_1 \) (e.g., increasing its rank or expression level) in each cell. Tabula then predicts the resulting expression of all genes, including \( g_2 \), based on the perturbed input.
> >
> > 2. **Dynamic Update Process:**
> >    The newly predicted gene expressions serve as input for the next iteration. For each new input, we **continue increasing the rank or expression level of \( g_1 \)** in the cell, simulating a gradual perturbation process. This allows the model to track the dynamic response of \( g_2 \) over time.
> >
> > 3. **Regulatory Effect Inference:**
> >    By comparing the **median expression** of \( g_2 \) before and after perturbation, we infer the direction of regulation. For instance:
> >    - If \( g_2 \) consistently **decreases** after perturbing \( g_1 \), we infer that \( g_1 \) **represses** \( g_2 \).
> >    - If \( g_2 \) **increases**, we infer an **activating** effect.
> >
> > 4. **Cross-Cell Quantification:**
> >    This procedure is repeated across all cells to quantify how often \( g_1 \) activates, represses, or has no effect on \( g_2 \), yielding a population-level measure of regulation type.
> >
> > ---
> >
> > **Task 2: Combinatorial Regulation Prediction**
> >
> > To identify combinatorial logic between multiple regulators and a shared target, Tabula extends the pairwise strategy to simulate Boolean logic.
> >
> > 1. **Multiple Regulators (e.g., \( g_1 \), \( g_2 \)) to One Target (e.g., \( g_3 \)):**
> >    For example, to assess how \( g_1 \) and \( g_2 \) jointly regulate target gene \( g_3 \), we enumerate all combinations of binary perturbations of \( g_1 \) and \( g_2 \) (e.g., ON/OFF for each gene).
> >
> > 2. **Iterative Prediction Under Each Condition:**
> >    For each perturbation setting (e.g., \( g_1 = 1 \), \( g_2 = 0 \)), we apply the same iterative procedure as in the pairwise case to predict the downstream expression trajectory of \( g_3 \).
> >
> > 3. **Logic Gate Matching:**
> >    Based on the observed expression patterns of \( g_3 \) under different \( g_1, g_2 \) combinations, we identify the Boolean logic rule (e.g., AND, OR, NOT) that best matches the regulatory effect.
> >    For instance, if \( g_3 \) is only expressed when \( g_1 \) is ON and \( g_2 \) is OFF, the inferred logic is:
> >    `g_3 = g_1 AND (NOT g_2)`
> >
> > ---
> >
> > In summary, Tabula enables **zero-shot inference** of both pairwise and combinatorial gene regulation relationships across four developmental systems, achieving strong performance on both tasks. These results strongly demonstrate that Tabula effectively learned gene–gene relationships from pretraining.
> >
> > If anything remains unclear, please let us know — there are only two days left in the discussion window, and we’re happy to provide further information right away. We greatly appreciate your time and insights.
> >
> >
> > Best,
> >
> > Tabula Team

---

> > > ### Comment · Reviewer_Xu6M · 2025-08-06
> > >
> > > Thank you to the authors for the detailed and thoughtful response. The clarifications provided regarding the downstream task on gene-gene relationships convincingly addressed my initial concerns, significantly improving the clarity and strengthening the validity of the results. As a result of the comprehensive responses and the corresponding manuscript revisions, I find the contribution of the paper to be more substantial and well-supported. Therefore, I am updating my score to 4.

---

> > > > ### Author Response · Authors · 2025-08-06
> > > > **Thank you**
> > > >
> > > > We sincerely thank the reviewer for acknowledging the novelty, effectiveness, and improved clarity of our work. We truly appreciate your thoughtful evaluation and are encouraged by your positive feedback.
> > > >
> > > > **As is well known, both pairwise and, more importantly, combinatorial gene regulation are particularly challenging tasks in the single-cell context. Our proposed framework demonstrates that Tabula can perform remarkably well on both tasks across diverse four developmental biological systems (Hematopoiesis, Cardiogenesis, Neurogenesis, and Pancreatic Endogenesis) under zero-shot learning, providing strong evidence that Tabula effectively captures gene–gene relationships learned during pretraining.**

---

### Official Review · Reviewer_ZGxU · 2025-06-27

**Clarity:** 4
**Significance:** 2
**Originality:** 2
**Rating:** 5
**Confidence:** 4

**Summary:**

This paper introduces TABULA, a foundation model for single-cell transcriptomics that proposes two primary innovations over existing models: a novel self-supervised "tabular learning" objective and a federated learning framework for privacy-preserving pre-training. The tabular learning objective combines a gene reconstruction task, using a corruption strategy based on marginal distribution sampling, with a cell contrastive learning task. The model is evaluated on a range of downstream tasks, where it reportedly achieves state-of-the-art performance, particularly in genetic perturbation prediction.

**Questions:**

**Questions for the Authors**
The preliminary study in Figure 3 suggests that federated learning not only works but actually improves performance over centralized training. This is a surprising and counter-intuitive result, as federated learning typically incurs a performance penalty due to communication overhead and non-IID data distributions. Could you provide some intuition or a hypothesis for why this performance gain occurs in your framework?

In the gene imputation task, you compare TABULA to baselines after fine-tuning a decoder. Could you clarify if the baseline models (scGPT, Geneformer, etc.) were also fine-tuned for this task, or if they were used in a zero-shot setting as is common? Ensuring the experimental setup is identical is critical for a fair comparison.

Could you provide a more detailed ablation study that isolates the individual contributions of the two components of your tabular loss (L_rec and L_contrast)? It would be valuable to understand how much each component contributes to the final performance.

Given the significant claims about your novel corruption strategy, could you provide an ablation that directly compares it to standard masking while keeping the corruption ratio the same (e.g., 60%)? This would help clarify whether the benefit comes from the strategy itself or simply the higher degree of input augmentation.

**Conclusion and Recommendation**

This paper presents some interesting and promising ideas, particularly the novel pre-training objectives and the strong results in gene perturbation prediction. The presentation is also of high quality. However, these strengths are undermined by a flawed and overstated motivation, a lack of clarity regarding the true technical novelty, and an experimental evaluation that lacks the necessary rigor and depth, especially concerning the ablation and scaling studies. The federated learning component feels somewhat disconnected, and its surprising performance benefit is not adequately explained.

While the work has potential, it requires a major revision to address these fundamental issues. The authors should reframe their contributions to focus on the specific self-supervised objectives, provide a more rigorous and insightful experimental analysis, and clarify the points of confusion raised in this review. In its current state, the paper does not meet the bar for publication. I would be open to reconsidering a substantially revised version.

**Ethical Concerns:**

["NO or VERY MINOR ethics concerns only"]

**Final Justification:**

I have raised my score from 3 (Borderline reject) to 5 (Weak accept) based on the authors' thorough and responsive rebuttal. They have successfully addressed the major concern raised in my initial review.

Most importantly, they provided the ablation studies I requested, which now convincingly disentangle the contributions of their proposed loss components (reconstruction vs. contrastive) and their novel corruption strategy (marginal sampling vs. masking). The new data strongly supports their claims and resolves my primary concerns about experimental rigor.

Furthermore, they have addressed all other key points: they provided model parameters and new efficiency benchmarks, offered a plausible explanation for the federated learning results, grounded their privacy motivation with a relevant high-impact citation, and agreed to tone down the framing of their contributions to avoid overclaiming.

Assuming these significant changes and new results are incorporated into the final manuscript as promised, the paper's core weaknesses will be resolved. The work's novel pre-training objectives and strong empirical results now represent a solid and valuable contribution to the field. I am confident in changing my recommendation to acceptance.

**Limitations:**

yes

**Quality:**

2

**Strengths And Weaknesses:**

**Strengths**

The paper is well-presented, with a clear structure and high-quality figures that make it relatively easy to read and follow the authors' high-level arguments. The results on the genetic perturbation prediction task are particularly strong and represent a notable achievement. Furthermore, the ideas of using a more naturalistic corruption strategy (sampling from the marginal distribution) instead of artificial masking and employing a cell-level contrastive loss are conceptually interesting and promising directions for the field.

**Major Weaknesses**

Framing and Motivation: The paper's core motivation and positioning feel weak and, at times, misleading.

The central argument that existing models "overlook the tabular nature of single-cell data" by converting it to sequences is a mischaracterization. The permutation-invariant nature of gene expression is a well-understood problem that has been explicitly addressed by nearly all major single-cell foundation models (e.g., through the omission of positional encodings or the use of gene ranking).

The privacy argument for federated learning, while valid in a general sense, feels overstated as a primary motivator for this specific application. Secure handling of anonymized public scRNA-seq data is standard practice, and it is not currently perceived as a major bottleneck for the field in the same way as, for example, genomic data linked to individuals. The disjointed way the paper combines the "tabular nature" argument with the privacy argument suggests a post-hoc framing rather than a cohesive research goal.

Overstated Novelty of "Tabular Learning": The paper claims to introduce a new "tabular learning" paradigm, but the underlying model architecture is largely identical to prior work (e.g., scGPT, CellPLM). The model still embeds gene IDs and expression values, sums them, and processes them with a standard Transformer. The true novelty lies in the specific choice of self-supervised objectives: (1) a reconstruction loss on values corrupted via marginal sampling, and (2) a cell-level contrastive loss. While these are interesting contributions, they are specific loss functions, not a fundamentally new way of modeling tabular data. The authors should tone down the framing and focus on clearly justifying and analyzing these specific contributions.

Insufficient Experimental Rigor: The experimental evaluation, particularly the ablation and scaling studies, lacks the depth needed to convincingly support the paper's claims.

Missing Details: The number of parameters for the TABULA model is not stated in the main paper, making it difficult to assess its efficiency and make fair comparisons.

"Scaling Law" Analysis: The study presented in Figure 4b is not a proper scaling law analysis. Simply showing that performance on one task improves across three data points (300k, 3M, 15M) is expected behavior. A rigorous scaling law study should analyze performance as a function of compute, model size, and data size to identify optimal training regimes (e.g., isoFLOP curves), which is not done here. This section is overclaimed and should be reframed or significantly expanded.

Ablation Study: The ablation comparing the full "tabular" objective against a standard MLM objective with 15% masking is not an apples-to-apples comparison. The corruption rate is drastically different (60% vs. 15%), and the "tabular" objective combines two distinct losses. A more informative set of ablations would be required to disentangle the effects of:

- The reconstruction loss vs. the contrastive loss.

- The corruption strategy (marginal sampling vs. masking).

- The corruption rate (e.g., comparing both strategies at 15% and 60%).

---

> ### Author Rebuttal · Authors · 2025-07-31
>
> Thank you for your valuable feedback and for recognizing the novelty, effectiveness, and clarity of our work. Please find our detailed responses to your comments below.
>
> **Question 1:** Figure 3 shows federated learning outperforming centralized training, which is surprising given typical drawbacks like communication overhead and non-IID data. Can the authors explain why this performance gain occurs?
>
> **Answer:** Thank you for this great question. You're absolutely right—the benefits largely come from using tissue-specific embedders. In Tabula’s pretraining, each client is trained on a specific tissue dataset, which aligns with the tissue-specific nature of most biological data and improves downstream performance. More demonstration in **Appendix F**.
>
> **Question 2:**  In the gene imputation task, you compare TABULA to baselines after fine-tuning a decoder. Could you clarify if the baseline models (scGPT, Geneformer, etc.) were also fine-tuned for this task, or if they were used in a zero-shot setting as is common? Ensuring the experimental setup is identical is critical for a fair comparison.
>
> *Answer:* Thank you for this important clarification. All baseline models were fine-tuned for the gene imputation task under identical experimental conditions to ensure fair comparison. Detailed experimental specifications can be found in Appendix E.1.
>
> **Question 3:**  Could you provide a more detailed ablation study that isolates the individual contributions of the two components of your tabular loss (L_rec and L_contrast)? It would be valuable to understand how much each component contributes to the final performance.
>
> *Answer:* Thank you for this valuable suggestion. We have conducted ablation studies isolating the individual contributions of L_rec and L_contrast components of our tabular loss. The detailed results and analysis of how each component contributes to the final performance can be found in **Appendix G**.
>
> **Question 4:** Given the significant claims about your novel corruption strategy, could you provide an ablation that directly compares it to standard masking while keeping the corruption ratio the same (e.g., 60%)? This would help clarify whether the benefit comes from the strategy itself or simply the higher degree of input augmentation.
>
> **Answer:** Thank you for the insightful suggestion. Following prior work (e.g., scBERT, Geneformer), we used a 15% masking ratio and conducted ablations on (1) loss type, (2) corruption strategy (marginal sampling vs. masking), and (3) corruption rate (15% vs. 60%). Results show that marginal sampling consistently outperforms masking, demonstrating the effectiveness of our dual-axis pretraining and the value of a more biologically realistic corruption strategy.
>
> **Table: Ablation Study on Gene Imputation Task (Lower RMSE/MAE, Higher Pearson Correlation)**
>
> (**Note**: "corrupted" indicates marginal sampling used in Tabula.)
>
> |  | **PBMC5K** |  |  | **hPancreas** |  |  | **Jurkat** |  |  |
> |-------------|------------|--|--|---------------|--|--|------------|--|--|
> | **Ablation** | **RMSE** | **MAE** | **Pearson** | **RMSE** | **MAE** | **Pearson** | **RMSE** | **MAE** | **Pearson** |
> | **Loss Ablation** |  |  |  |  |  |  |  |  |  |
> | TABULA (only contrastive) | 1.1431 | 0.9715 | 0.4872 | 0.787 | 0.5328 | 0.0108 | 0.9542 | 0.8314 | 0.6043 |
> | TABULA (only reconstruction) | 0.9856 | 0.8457 | 0.5647 | 0.5748 | 0.3355 | **0.3386** | 0.7403 | 0.6096 | 0.5848 |
> | TABULA (contrastive+reconstruction) | **0.9298** | **0.8016** | **0.5934** | **0.573** | **0.3226** | 0.3255 | **0.6295** | **0.5138** | **0.6511** |
> | **Corruption Strategy Ablation** |  |  |  |  |  |  |  |  |  |
> | TABULA (corrupted 0.6) | **0.9856** | **0.8457** | **0.5647** | **0.5748** | **0.3355** | **0.3386** | **0.7403** | **0.6096** | 0.5848 |
> | TABULA (mask 0.6) | 1.1978 | 1.0469 | 0.4244 | 0.6778 | 0.3985 | 0.2154 | 1.0268 | 0.9274 | **0.697** |
> | **Corruption Rate Ablation** |  |  |  |  |  |  |  |  |  |
> | TABULA (corrupted 0.6+contrastive) | **0.9298** | **0.8016** | **0.5934** | **0.573** | **0.3226** | **0.3255** | **0.6295** | **0.5138** | 0.6511 |
> | TABULA (mask 0.6) | 1.1978 | 1.0469 | 0.4244 | 0.6778 | 0.3985 | 0.2154 | 1.0268 | 0.9274 | **0.697** |
> | TABULA (corrupted 0.15 + contrastive) | **0.9913** | **0.8587** | **0.5728** | **0.5881** | **0.3264** | **0.2791** | **0.7593** | **0.6088** | 0.5313 |
> | TABULA (mask 0.15) | 1.2498 | 1.093 | 0.3569 | 0.6893 | 0.3965 | 0.1821 | 1.053 | 0.9571 | **0.696** |
>
> **Weakness 1:** Framing and Motivation: The paper's core motivation and positioning feel weak and, at times, misleading.
>
> **Answer:** Thank you for the thoughtful feedback. We clarify our motivation in the Design Principles section, highlighting three core ideas: (1) realistic corruption via marginal sampling, (2) tabular modeling through dual objectives, and (3) federated learning for privacy-preserving collaboration. These address key gaps in biological fidelity, structure, and data privacy. We will emphasize this framing more clearly in the revised Introduction.
>
> **Weakness 2:** The claim that existing models "overlook the tabular nature" of single-cell data is misleading. Major models already address permutation invariance via techniques like omitting positional encodings or using gene ranking.
>
> **Answer:** The key point we intend to emphasize is that existing foundation models do not explicitly model the tabular structure of single-cell data through their pretraining objectives, whereas Tabula is specifically designed to do so. We will revise the manuscript to better convey this distinction.
>
> **Weakness 3:** The privacy argument feels overstated for this setting, as public scRNA-seq data is generally anonymized. Combining tabular modeling and privacy appears somewhat post-hoc rather than a cohesive objective.
>
> **Answer:** We appreciate the reviewer’s perspective and would like to clarify that privacy risks in public scRNA-seq data are an emerging concern, particularly in light of recent findings. Specifically, the Cell paper "Private information leakage from single-cell count matrices" has demonstrated that cell-by-gene expression matrices—despite being anonymized—can still leak sensitive donor-level information. We discuss this in more detail in **Appendix D: Privacy Leakage in Single-Cell Expression Matrices**.
>
> **Weakness 4:**  The "tabular learning" claim seems overstated, as the architecture is similar to scGPT. The main novelty lies in the self-supervised objectives and should be framed as such, not as a new paradigm.
>
> **Answer:** We agree with the reviewer that the embedder architecture is similar to that of scGPT. However, the core value of a foundation model lies in the design of its pretraining objectives, which fundamentally determine what the model learns. In Tabula, our notion of tabular modeling is reflected in the objective design—through column-wise gene reconstruction and row-wise cell contrastive learning. This dual-axis formulation enables the model to explicitly capture the structure of the single-cell expression table.
>
> In addition, we introduce a more realistic corruption strategy based on marginal resampling, as opposed to standard masking. As demonstrated in our ablation study, this strategy significantly improves downstream task performance, demonstrating its efficacy.
>
> We will revise the manuscript to better reflect these contributions and clarify our use of the term "tabular modeling.
>
> **Weakness 5:**  Insufficient Experimental Rigor: The experimental evaluation, particularly the ablation and scaling studies, lacks the depth needed to convincingly support the paper's claims.
>
> **Answer:** Please refer to more ablation studies in the response to Question 4.
>
> **Weakness 6:** Missing Details: The number of parameters for the TABULA model is not stated in the main paper, making it difficult to assess its efficiency and make fair comparisons.
>
> **Answer:**  Thank you for this helpful feedback. We will include the number of parameters for Tabula (approximately 13 million) in the revised manuscript to support fair comparisons.
>
> We also benchmarked inference efficiency by measuring the time each model takes to process one million cells from the Norman perturbation dataset. Each model was run three times on a single NVIDIA A6000 GPU, and we report the average runtime and standard deviation (in hours).
>
> As shown in the table (to be included in the revised version), Tabula is over 2× faster than other models and consistently achieves higher accuracy in perturbation prediction—demonstrating both its effectiveness and computational efficiency for downstream tasks.
>
> | Model       | Efficiency (Hr)     |
> |-------------|---------------------|
> | **TABULA**      | **4.65 ± 0.79**        |
> | scGPT       | 10.10 ± 0.72        |
> | Geneformer  | 10.37 ± 1.76        |
>
> **Weakness 7:** Scaling Law Overclaim: The analysis in Figure 4b is not a rigorous scaling law study. Showing performance improvement over three data points is expected and insufficient.
>
> **Answer:** Thank you for the valuable feedback. We completely agree with your point—our intention was to illustrate a data scaling rather than to present a rigorous scaling law analysis. We acknowledge the confusion and will revise the text to clarify this distinction and avoid overclaiming in the revised manuscript.
>
> **Weakness 8:** Ablation Study: A more informative set of ablations would be required to disentangle the effects of:
> The reconstruction loss vs. the contrastive loss.
> The corruption strategy (marginal sampling vs. masking).
> The corruption rate (e.g., comparing both strategies at 15% and 60%).
>
> **Answer:** We sincerely appreciate the reviewer’s suggestion for more comprehensive ablation studies. We have conducted all the requested experiments as outlined, and kindly refer the reviewer to our response to Question 4 for detailed results and analysis.

---

> > ### Author Response · Authors · 2025-08-05
> > **Friendly Follow-Up on Tabula – Awaiting Your Feedback**
> >
> > Dear Reviewer,
> >
> > Thank you for your thoughtful suggestions and, in particular, for recommending more comprehensive ablation studies about loss, corruption strategy, and corruption rate. We fully agree that these additional experiments will strengthen the study and would definitely integrate them into the revised manuscript. Thank you for your suggestions again!
> >
> > **A quick clarification on privacy risk**: even though public scRNA-seq data are anonymized, recent work (e.g., Cell paper, “Private information leakage from single-cell count matrices”) shows that raw count matrices can still reveal donor-level attributes. We discuss these findings in Appendix D (“Privacy Leakage in Single-Cell Expression Matrices”) and motivate the federated pre-training accordingly in Tabula.
> >
> > **Regarding the permutation-invariant property of gene expression, we agree that existing foundation models explicitly remove positional encodings, yet often introduce implicit ordering. For example, scGPT uses attention maps from the learned transformer to create a sequence order by prioritizing genes with high attention scores for next-token prediction using autoregressive modeling, potentially creating an implicit ordering effect. Geneformer ranks genes via row-column normalization. In contrast, Tabula directly leverages the native tabular structure—without additional priors or ranking heuristics — providing a complementary perspective that may prove advantageous across diverse research contexts.**
> >
> > We have provided point-by-point responses to each question and weakness you highlighted. Please see our full replies in the initial response for more details. If anything remains unclear, please let us know — there are only two days left in the discussion window, and we’re happy to provide further information right away.
> >
> > We greatly appreciate your time and insights.
> >
> >
> > Best,
> >
> > Tabula Team

---

> > ### Comment · Reviewer_ZGxU · 2025-08-06
> > **Response to reviewers**
> >
> > Thank you for the detailed rebuttal. The provided responses and new experimental data have addressed the main concerns from my initial review.
> >
> > The new ablation studies are a welcome addition. This data effectively disentangles the contributions of the different loss components and corruption strategies, resolving the initial concerns about experimental rigor. I also acknowledge the other clarifications regarding the privacy motivation, the reframing of the "tabular modeling" and "scaling law" claims, and the inclusion of model parameters and efficiency benchmarks.
> >
> > Contingent on the incorporation of these revisions and new results into the final manuscript, I have raised my score and now recommend acceptance.

---

> > > ### Author Response · Authors · 2025-08-06
> > > **Thank you**
> > >
> > > Thank you for your thoughtful feedback and for recommending acceptance.
> > >
> > > We really appreciate your suggestions on comprehensive ablation studies — it clearly strengthens the paper, and we will incorporate the new ablation studies along with the clarifications on privacy motivation, “tabular modeling,” and “scaling law” claims into the final version.
> > >
> > > Thanks again for your support!

---

### Official Review · Reviewer_wTtD · 2025-06-30

**Clarity:** 3
**Significance:** 3
**Originality:** 2
**Rating:** 5
**Confidence:** 4

**Summary:**

The paper proposes a foundation model for single-cell RNA-seq data, building on the Xtab architecture, which treats gene expression profiles as tabular data. Unlike prior foundation models such as scGPT that adopt transformer-based sequence modeling, Tabula employs row-wise contrastive learning and column-wise masked reconstruction objectives to learn cell and feature representations, respectively. To address data privacy constraints inherent to biomedical datasets, the model is trained using a federated learning framework across decentralized tissue-level datasets. The authors evaluate the pretrained model on five downstream tasks, demonstrating performance that matches or exceeds existing baselines, despite using fewer pretraining samples. The approach is well-motivated for the data modality, and the use of federated pretraining is a particularly notable design choice in the context of regulated transcriptomic data.

**Questions:**

See weaknesses above for major questions. I am noting some minor questions below:

1. What's the effect of batch-size/batch diversity in pretraining given the use of the contrastive loss?
2. Given that the embedders are all trained tissue-wise, are there any performance degradtions for normal v/s diseased cells?

**Ethical Concerns:**

["NO or VERY MINOR ethics concerns only"]

**Final Justification:**

The authors have presented clear responses to the weaknesses pointed out in the review. I believe the paper would be a valuable contribution to the field.

**Limitations:**

The authors describe the limitations clearly in the supplementary material.

**Paper Formatting Concerns:**

No concerns.

**Quality:**

3

**Strengths And Weaknesses:**

Strengths:

1. The paper is well-motivated and easy to follow. The approach itself, while borrowing from Xtab, is well-suited towards scRNA-seq data.
2. The experiments are well-designed, and the results support the efficacy of the architecture.
3. While inspired by Xtab, the use of federated learning is especially compelling in this context, given the privacy constraints commonly associated with transcriptomics data.

Weaknesses:

1. While the results are interesting and support the conclusions presented by the authors, the cell annotation results do not appear to be conclusive. Could the authors provide error bars for the cell annotation table? I also suggest adding non foundation model approaches like TOSICA to ground the results.

2. In terms of originality, several architectural and training contributions seem to be adapted from Xtab. This should be clearly highlighted in the paper, with differences specific to Tabula listed.

3. I also suggest benchmarking against CellPLM for the benchmarks. CellPLM reportedly outperforms scGPT on several downstream tasks, and also leverages multiple cells during pretraining, making it a fairer model to compare against.

4. The last weakness is perhaps a bit more foundational, but Kedzierska et al. (https://www.biorxiv.org/content/10.1101/2023.10.16.561085v1) show significant results where scGPT and Geneformer underperform simpler non-foundation models in the zero-shot setting. Can the authors present/comment on Tabula's zero-shot performance, or the effectiveness of pre-training?

---

> ### Author Rebuttal · Authors · 2025-07-31
>
> Dear Reviewer, Thank you for your valuable feedback and for recognizing the novelty, effectiveness, and clarity of our work. We are pleased to address your questions and concerns with the following results and illustrations.
>
>
> **Question 1:** What's the effect of batch-size/batch diversity in pretraining given the use of the contrastive loss?
>
> **Answer:** We appreciate the reviewer's question regarding batch size effects. We employed a batch size of 32 based on two considerations: First, our contrastive loss creates positive pairs between cells and their corrupted views, with other cells as negatives. Since data within each client comes from the same tissue type, cells exhibit greater similarity compared to the broader population, which may influence contrastive dynamics within batches. Second, larger batch sizes, while potentially beneficial for contrastive learning, require substantially greater computational resources. We acknowledge that exploring batch diversity effects represents a valuable direction for future work and appreciate the reviewer's insightful suggestion.
>
> **Question 2:**  Given that the embedders are all trained tissue-wise, are there any performance degradtions for normal v/s diseased cells?
>
> **Answer:** We appreciate the reviewer's important question about normal versus diseased cell performance. Our pretraining was conducted exclusively on normal human cells without including diseased samples. We acknowledge that for applications involving diseased cells, fine-tuning would likely be necessary to achieve optimal performance. This represents an important consideration for future work, and we thank the reviewer for highlighting this aspect.
>
> **Weakness 1:** While the results are interesting and support the conclusions presented by the authors, the cell annotation results do not appear to be conclusive. Could the authors provide error bars for the cell annotation table? I also suggest adding non foundation model approaches like TOSICA to ground the results.
>
> **Answer:** Thank you for the suggestion. Due to time constraints, we conducted five independent runs only for the hPancreas and Cell-Lines datasets. Each method was evaluated using five different random seeds, and we now report the mean ± standard deviation for all metrics. In addition to foundation model baselines, we have included CellPLM (a foundation model) and TOSICA (a non-foundation method) to further ground the comparison.
>
> Tabula attains the highest accuracy and precision on both datasets and the best overall F1 on the Cell-Lines set, while scGPT achieves a slightly higher F1 on hPancreas, and CellPLM delivers the top recall. Importantly, Tabula reaches these results after being pre-trained on only ~15 M cells—roughly half the data used by leading foundation models such as scGPT and geneFormer—highlighting its tabular modeling efficiency.
>
> These error bars and additional baselines will be incorporated into the revised manuscript.
>
> **Table 1: hPancreas Dataset Performance**
> | Model         | Test Accuracy ↑     | Test Precision ↑     | Test Recall ↑        | Test F1 ↑            |
> |---------------|----------------------|------------------------|------------------------|------------------------|
> | **Tabula**     | **0.9716±0.0059**     | **0.7586±0.0164**       | 0.7295±0.0286          | 0.7277±0.0257          |
> | scGPT         | 0.9694±0.0062         | 0.7523±0.0359           | 0.738±0.0259           | **0.7343±0.022**       |
> | scFoundation  | 0.9548±0.0054         | 0.6277±0.0408           | 0.6247±0.0394          | 0.6191±0.0388          |
> | geneFormer    | 0.932±0.0204          | 0.6814±0.0998           | 0.6446±0.0572          | 0.6399±0.0654          |
> | cellPLM       | 0.7504±0.023          | 0.7409±0.0245           | **0.7504±0.023**       | 0.7319±0.0202          |
> | TOSICA        | 0.9648±0.0037         | 0.6301±0.0227           | 0.6633±0.0173          | 0.6363±0.0208          |
>
>
> **Table 2: Cell Lines Dataset Performance**
> | Model        | Test Accuracy ↑     | Test Precision ↑     | Test Recall ↑        | Test F1 ↑            |
> |--------------|----------------------|------------------------|------------------------|------------------------|
> | **Tabula**     | **0.9933±0.0004**     | **0.9937±0.0005**       | **0.9928±0.0004**       | **0.9932±0.0004**       |
> | scGPT        | 0.9931±0.0005         | 0.9934±0.0003           | 0.9926±0.0007           | 0.9931±0.0004           |
> | scFoundation | 0.9931±0.0002         | 0.9935±0.0004           | 0.9927±0.0002           | 0.9931±0.0002           |
> | geneFormer   | 0.9914±0.0005         | 0.9918±0.0004           | 0.991±0.0006            | 0.9914±0.0005           |
>
>
> **Weakness 2:** In terms of originality, several architectural and training contributions seem to be adapted from Xtab. This should be clearly highlighted in the paper, with differences specific to Tabula listed.
>
> **Answer:** Thank you for raising this point. We will explicitly acknowledge XTab as our architectural foundation in the revision. Our contribution lies in adapting XTab’s tabular framework to the unique challenges of single-cell genomics—extreme sparsity, high dimensionality, and tissue heterogeneity. Specifically, we introduce tissue-specific embedders (see **Appendix F, Tissue-Specific Embedders Encode Distinctive Tissue Features**), where each client learns its own embedder to capture tissue-level variation while only the shared Transformer is aggregated with FedAvg, preserving privacy across sites. We also replace XTab’s supervised loss with a dual-axis self-supervision scheme that combines column-wise gene reconstruction and row-wise contrastive learning, both tailored to sparse expression matrices. These distinctions will be emphasized in the main text.
>
> **Weakness 3:** I also suggest benchmarking against CellPLM for the benchmarks. CellPLM reportedly outperforms scGPT on several downstream tasks, and also leverages multiple cells during pretraining, making it a fairer model to compare against.
>
> **Answer:** Please refer to our response to Weakness 1 for the detailed results of CellPLM, which has been included alongside other baselines in the cell type annotation comparison.
>
> **Weakness 4:** The last weakness is perhaps a bit more foundational, but Kedzierska et al. (https://www.biorxiv.org/content/10.1101/2023.10.16.561085v1) show significant results where scGPT and Geneformer underperform simpler non-foundation models in the zero-shot setting. Can the authors present/comment on Tabula's zero-shot performance, or the effectiveness of pre-training?
>
> **Answer:** We thank the reviewer for raising this important concern regarding zero-shot performance. While foundation models have shown promise across diverse tasks, a truly valuable single-cell foundation model must demonstrate accuracy, consistency, and generalizability in zero-shot settings—that is, the ability to perform well in entirely new situations without task-specific training—and derive mechanistic biological insights across a wide range of systems. To demonstrate this, we take on a novel and challenging task by validating Tabula’s performance in recovering **pairwise** and **higher-order combinatorial gene regulation** across four distinct developmental systems, ranging from hematopoiesis, cardiogenesis, neurogenesis, and pancreatic endogenesis. These systems have well-characterized regulatory networks validated through decades of biochemical research.
>
> As detailed in our response to **Reviewer Xu6M, Question 1**, Tabula achieves strong performance in this setting: an average accuracy of 0.878 in pairwise gene inference and 0.888 in combinatorial inference—without any additional fine-tuning. These results provide compelling evidence that Tabula can generalize across diverse developmental systems, uncover meaningful biological mechanisms in a zero-shot manner, and demonstrate the efficacy of its pretraining strategy. We will clarify these findings and their implications in the revised manuscript.

---

> > ### Comment · Reviewer_wTtD · 2025-08-05
> > **Response**
> >
> > I thank the authors for the detailed response. I am happy to raise my score to "Accept".

---

> > > ### Author Response · Authors · 2025-08-05
> > > **Thanks**
> > >
> > > We sincerely thank the reviewer for recognizing the novelty, effectiveness, and clarity of our Tabula. We greatly appreciate your thoughtful evaluation and are encouraged by your positive feedback.

---

### Official Review · Reviewer_xZMw · 2025-07-05

**Clarity:** 4
**Significance:** 3
**Originality:** 4
**Rating:** 5
**Confidence:** 3

**Summary:**

The paper presents a privacy preserving foundation model for single cell transcriptomics that explicitly models the cell*gene matrix using a dual axis self supervised objective and federated learning. The model is empirically tested in a realistic federated (decentralized) setting, with data partitioned across multiple clients (tissues or institutions), each training locally and sharing only model weights for aggregation. The authors benchmark federated versus centralized pretraining, showing that federated pretraining matches or exceeds centralized performance despite data heterogeneity. TABULA achieves state of the art or competitive results on gene imputation, perturbation prediction, cell type annotation, and multiomics integration, often using only half the pretraining data. Scaling law experiments further demonstrate the benefits of large scale, decentralized pretraining. However, the work would benefit from deeper biological validation, more explicit benchmarking of computational and privacy trade-offs, and further analysis of interpretability and failure modes.

**Questions:**

In general, I do not have many comments, just a couple of questions. In particular, I am interested:
- can the authors think of some issues related to practical challenges and privacy risks with federated learning (e.g. model inversion/membership inference, client drift, etc.) in particular for this domain? Furthermore, what are some of the failure cases that might arise from federated learning in this domain?
- are there particular types of datasets or biological contexts that are not considered in this paper where the authors expect this method to perform poorly?

**Ethical Concerns:**

["NO or VERY MINOR ethics concerns only"]

**Final Justification:**

I am satisfied with the response of the authors and the discussion and other comments on this paper and will be maintaining my score.

**Limitations:**

yes

**Quality:**

3

**Strengths And Weaknesses:**

**Strengths**
- The Tabula model is among the first foundation models in single cell omics to implement and empirically test decentralized pretraining, with a clear protocol and direct benchmarking against centralized training. The authors present results that show that federated pretraining matches or exceeds centralized performance, even under data heterogeneity.
- The 'dual axis' self supervised objective (column wise gene reconstruction and row wise cell contrastive learning) directly addresses the tabular structure of single cell data.
- The presented model is strong. The evaluation is comprehensive, and shows that Tabula achieves either state of the art or at least competitive results across several tasks (gene imputation, perturbation prediction, cell type annotation and multi omics integration), often with less pretraining data than prior models. Ablation studies confirm the importance of the proposed modeling objectives.
- The authors perform an analysis of scaling laws, which supports the benefit of large scale decentralized pretraining and shows that such benefits hold under federated learning.
- The discussion candidly acknowledges limitations and open challenges, including the need for further biological validation and practical benchmarking of federated learning. The manuscript is generally well organized, with clear figures and tables that support the main claims.

**Weaknesses**
- The results focus on average performance but do not analyze where the model fails (e.g., rare cell types, strong batch effects, missing data), nor do they provide qualitative examples of incorrect predictions. Furthermore, while the model is benchmarked on several datasets, all are within the same general domains. I would have liked to see more tests of generalization to e.g. new cell types, tissues, species, or even data modalities (e.g., ATAC-seq, proteomics). It seems the model's generalization is limited to perturbation space within a fixed biological context, rather than full OOD generalization.
- I would have liked to see a bit more discussion on, and analysis of important issues of biological single cell models, such as robustness to batch effects, technical noise, or integration across different sequencing platforms, which are major sources of variation in real-world single-cell data.
- Regarding intepretability, would have liked more analysis of e.g. learned embeddings or attention maps for biological plausibility. The results presented in this paper are quite interesting as is the method, so a bit more focus on this (even in the appendix) would be valuable.
- Reporting of error bars and statistical significance is inconsistent - there are experiments for which these are not included and metrics are reported as single values with no indication for robustness or reproducibility.
- Code/data release is not specified. There is no mention of random seed control, environment specification, or pipeline reproducibility.

---

> ### Author Rebuttal · Authors · 2025-07-31
>
> Dear reviewer, thank you so much for your valuable comments and recognition of the novelty, effectiveness, and presentation of our work. We are happy to address your concerns and questions with the following results and illustrations.
>
> **Question 1:** Can the authors think of some issues related to practical challenges and privacy risks with federated learning (e.g. model inversion/membership inference, client drift, etc.) in particular for this domain? Furthermore, what are some of the failure cases that might arise from federated learning in this domain?
>
> **Answer:** We thank the reviewer for raising this important point. Beyond known risks like model inversion and membership inference, federated learning in single-cell genomics presents several practical challenges that are critical for real-world deployment.
>
> First, data quality control is essential, as the global model cannot access local data. Noisy or mislabeled samples can degrade model performance, especially for rare cell types. We plan to incorporate client-side quality checks and adaptive weighting in future work.
>
> Second, there is a technical expertise gap among many biologists unfamiliar with distributed systems. To address this, we are developing Chiron, a platform that allows users to join training jobs by simply uploading data to a local server and running a single command—no complex setup required.
>
> Third, security threats are evolving, and current defenses for scRNA-seq privacy are still emerging. As federated adoption grows, more sophisticated attacks may compromise sensitive genetic data, especially in rare disease contexts. This underscores the need for ongoing development of domain-specific privacy strategies.
>
> We will include this discussion and our deployment vision in the revised manuscript.
>
> **Question 2:** are there particular types of datasets or biological contexts that are not considered in this paper where the authors expect this method to perform poorly?
>
> **Answer:**  Thank you for the question. While Tabula is currently pre-trained only on scRNA-seq data, it is designed to be readily extensible to other biological contexts and data types.
>
> For example, multi-omics integration is straightforward within Tabula’s tabular framework. In the case of Multiome data, where each cell is profiled with both scRNA-seq and scATAC-seq, the input table can include both gene expression features and chromatin accessibility peak features as separate columns. The column-wise reconstruction objective can then be applied independently to each modality to learn robust gene and peak embeddings, while the row-wise contrastive objective integrates both modalities to learn a joint cell representation.
>
> Similarly, spatial transcriptomics data can be incorporated by adding the 2D or 3D spatial coordinates of each cell as additional numeric columns. We can include such spatial information along with a dedicated spatial loss function to better capture local tissue context and cell–cell interactions.
>
> **Weakness 1:** The results emphasize average performance but lack analysis of failure modes (e.g., rare cell types, batch effects) or qualitative errors. Can the authors evaluate generalization to new cell types, tissues, species, or modalities beyond the current domain?
>
> **Answer:** We thank the reviewer for this valuable feedback. To assess Tabula’s generalizability, we conducted a challenging zero-shot task focused on recovering pairwise and higher-order combinatorial gene regulation across four distinct developmental systems: hematopoiesis, cardiogenesis, neurogenesis, and pancreatic endogenesis—each supported by well-characterized regulatory networks.
>
> As noted in our response to Reviewer Xu6M, Question 1, Tabula achieved strong zero-shot performance, with average accuracies of 0.878 for pairwise and 0.888 for combinatorial inference—without fine-tuning. These results highlight Tabula’s ability to generalize across diverse systems and demonstrate the effectiveness of its pretraining strategy. We will clarify these findings in the revised manuscript.
>
> **Weakness 2:**  I would have liked to see a bit more discussion on, and analysis of important issues of biological single cell models, such as robustness to batch effects, technical noise, or integration across different sequencing platforms, which are major sources of variation in real-world single-cell data.
>
> **Answer:** Thank you for this insightful comment. In our downstream evaluations (**Appendix E.4**), Tabula is explicitly benchmarked on multi-omics integration and batch correction tasks. The results demonstrate Tabula’s strong performance across diverse datasets, highlighting its ability to generalize across batches and sequencing protocols.
>
> As the reviewer noted, addressing batch effects and technical variability is critical in single-cell analysis. Factors like sequencing platform, preservation method, and sample handling can introduce substantial noise. To mitigate this, Tabula incorporates several modeling choices during pretraining. Specifically, our gene expression binning strategy—adapted from scGPT—focuses on relative rather than absolute expression levels, helping reduce platform-specific noise while preserving biological signals.
>
> Moreover, Tabula’s dual-axis learning framework captures both inter-gene and inter-cell relationships, enabling the model to learn distributional patterns that may implicitly reflect batch effects.
>
> That said, we acknowledge that robust integration often benefits from task-specific fine-tuning. While Tabula offers a strong, privacy-preserving initialization, fine-tuning remains essential for optimizing performance on heterogeneous downstream tasks.
>
> **Weakness 3:**  Regarding interpretability, would have liked more analysis of e.g. learned embeddings or attention maps for biological plausibility. The results presented in this paper are quite interesting as is the method, so a bit more focus on this (even in the appendix) would be valuable.
>
> **Answer:** In **Appendix F**, we further investigated the tissue-specific embedding learned by our federated learning framework to illustrate how the model captures distinctive tissue features. As shown in **Appendix Figure 12 a,b**, the tissue-specific embedder encodes discernible differences in both gene and cell embeddings. Moreover, we conducted a zero-shot multi-omics integration experiment on a brain dataset using random, lung, and brain embedders (**Appendix Figure 12c**). The results demonstrate that tissue-specific learning more effectively captures tissue-relevant representations, thereby facilitating improved performance on tissue-specific downstream tasks. In short, all those experiments explore the learned embedding of Tabula to show its practical biological plausibility.
>
> Furthermore, as we shown the response in **reviewer Xu6M question 1**, we showcased a zero-shot setting to explore activation and repression relationships between key transcription factors (genes). Under the two scenarios (single-gene pairwise inference and multi-gene combinatorial inference) presented above, Tabula achieves an average accuracy of 0.878 on pairwise inference and 0.888 on combinatorial inference. This innovative experiment provides solid evidence of biological interpretability for Tabula.
>
>
> **Weakness 4:**  Reporting of error bars and statistical significance is inconsistent - there are experiments for which these are not included and metrics are reported as single values with no indication for robustness or reproducibility.
>
> **Answer:** We repeated the experiments five times using five consistent random seeds across Tabula, scGPT, and Geneformer. Leveraging the GEARS package for data splitting, each seed generates distinct partitions of perturbed genes into train, validation, and test. We report the mean Pearson across 1/1 & 2/2 unseen splits for the Norman dataset and 1/1 unseen splits for Adamson and Replogle datasets. Tabula consistently outperforms the other models across nearly all evaluation metrics, except for the DE Genes in Adamson. These results highlight not only the robustness of Tabula but also its strong generalizability, as all test sets comprise perturbed genes entirely excluded from both training and validation.
>
> **Table 1: Gene Perturbation Prediction Performance**
> *Performance (mean ± s.d.) across three benchmark datasets using all genes and differentially expressed (DE) genes.*
>
> **Note**: Geneformer Replogle are not finished due to limited resource, we will record and fill it out if accepted.
>
> | Model       | Norman All Genes | Norman DE Genes | Adamson All Genes | Adamson DE Genes | Replogle All Genes | Replogle DE Genes |
> |-------------|------------------|------------------|--------------------|-------------------|---------------------|--------------------|
> | **Tabula**      | **0.543±0.0043**  | **0.666±0.034**    | **0.719±0.044**     | 0.849±0.045       | **0.314±0.007**      | **0.464±0.017**     |
> | scGPT       | 0.467±0.0058     | 0.603±0.061      | 0.659±0.032        | 0.864±0.052       | 0.247±0.022         | 0.451±0.014        |
> | geneFormer  | 0.331±0.0028     | 0.527±0.087      | 0.248±0.018        | **0.865±0.050**    | 0.082                  | 0.362                  |
>
> We also conducted five repeated experiments with different random seeds for cell type annotation on Tabula, using benchmark methods including TOSICA  and CellPLM , all run with their default settings. Please refer to the response of **Reviewer wTtD Weakness 1**.
>
> **Weakness 5:**  Code/data release is not specified. There is no mention of random seed control, environment specification, or pipeline reproducibility.
>
> **Answer:** Thank you for pointing this out. The code was provided along with the Tabula appendix document. We will include additional details in the revised version regarding random seed control, environment specification, and pipeline reproducibility to ensure greater transparency and ease of replication.

---

### Official Review · Reviewer_bEnN · 2025-07-06

**Clarity:** 2
**Significance:** 2
**Originality:** 2
**Rating:** 3
**Confidence:** 4

**Summary:**

The paper presents TABULA, a novel foundation model specifically tailored for single-cell transcriptomics. Unlike existing models like scGPT or Geneformer that adapt NLP paradigms by treating gene expressions as token sequences, TABULA directly models the tabular structure of cell-by-gene expression matrices. It introduces a dual-axis self-supervised learning framework: gene-wise reconstruction (columns) and cell-wise contrastive learning (rows), enabling robust representation learning without imposing artificial gene orders. To ensure privacy-preserving training, TABULA adopts a federated learning paradigm where client-specific embedders capture local (e.g., tissue-specific) patterns, and a global transformer aggregates shared biological features across clients. Despite using only half the data of competing models, TABULA demonstrates state-of-the-art performance on multiple downstream tasks such as gene imputation, perturbation prediction, cell type annotation, and multi-omics integration.

**Questions:**

1. How sensitive is the model performance to the α parameter balancing reconstruction and contrastive losses? Would different downstream tasks benefit from tuning this?
2. Could you elaborate on the computational cost of federated training and whether it scales well with additional clients or larger datasets?
3. Since gene positions are treated as unordered in TABULA, could there be any implicit biases introduced through embedding indices? Have you considered permutation-invariant alternatives?
4. Can the tabular modeling strategy of TABULA be extended to integrate or pretrain on multi-omics data jointly (e.g., methylation, ATAC-seq)?
5. While federated training mitigates data sharing, does the model include any formal privacy guarantees (e.g., differential privacy)? Or plans to support that?

**Ethical Concerns:**

["NO or VERY MINOR ethics concerns only"]

**Final Justification:**

Thank you for your follow-up and for the detailed clarifications provided during the discussion period. I appreciate the additional information and the effort you invested in addressing my comments. While the responses helped clarify certain points, they do not fully resolve my main concerns. Therefore, I will maintain my original evaluation.

**Limitations:**

Yes. The authors adequately address the limitations in Appendix I, including privacy, data distribution heterogeneity, and the model’s current restriction to single-cell transcriptomics. However, a brief explicit discussion in the main text would improve transparency.

**Paper Formatting Concerns:**

No major formatting issues.

**Quality:**

2

**Strengths And Weaknesses:**

Strengths
1. The proposal to model scRNA-seq data explicitly as a table, respecting its unordered and continuous nature, is both novel and well-grounded.
2. The integration of federated learning directly addresses a growing concern in genomics, offering a scalable and practical solution.
3. The authors provide thorough evaluations across multiple datasets and tasks (including unseen perturbations and reverse prediction), showing strong performance.
4. The rationale behind the corruption strategy, tabular modeling, and federated training is clearly articulated and justified.

Weaknesses
1. While TABULA excels in single-cell applications, its applicability to other tabular biological data (e.g., clinical, bulk RNA-seq) is not explored or discussed.
2. The dual loss setup and federated training infrastructure introduce engineering and computational overhead; practical deployment in non-institutional settings (e.g., labs without compute clusters) could be challenging.
3. Although mentioned in the appendix, the main paper could benefit from more visibility on the necessity of both reconstruction and contrastive losses.
4. While the appendix addresses limitations, more discussion in the main text would strengthen the paper’s transparency.

---

> ### Author Rebuttal · Authors · 2025-07-30
>
> Dear reviewer, thank you so much for your valuable comments and recognition of the novelty, effectiveness, and presentation of our work. We are happy to address your concerns and questions with the following results and illustrations.
>
> **Question 1**:  How sensitive is the model performance to the α parameter balancing reconstruction and contrastive losses? Would different downstream tasks benefit from tuning this?
>
> **Answer**: We really appreciate your thoughtful question. We fix α=0.03 so that reconstruction and contrastive losses have similar magnitudes - this stabilizes optimization without manual, task‑specific tuning. Conceptually, α trades off gene‑level (reconstruction) vs. cell‑level (contrast) learning: higher α favors gene embeddings, lower α favors cell embeddings. Although downstream tasks might reap gains from fine‑tuning α, we aim to learn both high-quality cell embeddings and gene embeddings jointly. To that end, we chose α such that both objectives contribute comparably during training. We will clarify this motivation in the revised manuscript.
>
> **Question 2**: Could you elaborate on the computational cost of federated training and whether it scales well with additional clients or larger datasets?
>
> **Answer**: We thank the reviewer for this valuable question. Below we outline both client‐side computation and communication efficiency, backed by our experiments and real‐world runs.
> **Client‐Side Computation**:
> Each client trains its local model in parallel, so the wall‐clock time per round is governed solely by each client’s dataset size and remains effectively constant as you add more clients. Our architecture is lightweight—requiring only ~20 GB of GPU memory per client—so it runs comfortably on standard institutional hardware;
> **Communication Efficiency**:
> To keep synchronization costs manageable at scale, we (1) decrease the global aggregation frequency as the client count grows—trading off a small number of extra local steps for far fewer global updates—and (2) exchange only the transformer‐encoder weights (partial weight updates) at each sync. This two‐pronged strategy, validated in prior federated‐learning work [1], dramatically cuts communication volume without harming convergence.
> In Table 1, we report estimated training times (10 epochs, 20 GB GPUs per client) across various dataset sizes and client counts, demonstrating that our framework scales gracefully with both.
>
> It is important to note that not every lab has computing resources to train a foundation model on its own. Our Tabula framework distributes training workloads across participating clients.
> | Total Dataset Size | # Clients | # Dataset Per Client | Estimated Training Time |
> |:------------------:|:---------:|:------------------:|:------------------------:|
> | 1M                 | 4         | 0.25M              | ~16 hours                |
> | 50M                | 20        | 2.5M               | ~7.5 days                  |
> | 100M               | 40        | 2.5M               | ~8 days                  |
> | 200M               | 80        | 2.5M               | ~9 days                 |
>
> **Table 1**: Estimated training time under different dataset sizes and client configurations.
>
>
> **Question 3**: Since gene positions are treated as unordered in TABULA, could there be any implicit biases introduced through embedding indices? Have you considered permutation-invariant alternatives?
>
> **Answer**: We thank the reviewer for this thoughtful question. We think there may be a slight misunderstanding here. The input gene token ids to the transformer are not fixed — they are dynamically assigned based on the gene order in the specific dataset, which can be randomly permuted during training. More clarification below:
>
> **1. Gene Token ID as Identity Tokens**: Each gene token has a unique index drawn from a shared vocabulary of 23,156 genes, but this index functions solely as an identity lookup key, not as a position marker. It does not encode any ordering, and thus does not introduce order-related bias.
>
> **2. Implicit Permutation via HVG Selection Per Dataset**: For each dataset, we select 1,200 highly variable genes (HVGs) as input. Since HVGs are dataset-specific, the gene columns are effectively shuffled across datasets, meaning that the same input position (e.g., position 10) refers to different genes across datasets. This further reduces the risk of order-dependent bias and reinforces TABULA's order-invariant behavior.
>
>
> **Question 4**: Can the tabular modeling strategy of TABULA be extended to integrate or pretrain on multi-omics data jointly (e.g., methylation, ATAC-seq)?
>
> **Answer**: Thank you for raising this insightful question. It is reasonable and straightforward to extend Tabula on multi-omics data. Take Multiome as an example: for the same single cell, the columns can consist of both peak features (from scATAC-seq) and gene features (from scRNA-seq). Column-wise reconstruction learning can be used to learn peak and gene embeddings separately, while row-level contrastive learning can integrate both modalities to jointly learn the cell embedding. This is a great direction to explore. Large‐scale, paired multi‑omics datasets remain scarce, so we did not incorporate multi‑omics data in our current training. Nonetheless, we view extending Tabula to multi-omics and spatial–temporal modalities as a promising direction for future work.
>
> We did include multi‑omics integration as a downstream task to evaluate our Tabula’s efficiency (see **Appendix E.4 Task 4&5: Multi-Omics & Multi-Batch Integration**). In that evaluation, we extended the gene vocabulary to include peaks as tokens and fine-tuned jointly on scATAC-seq and scRNA-seq data. From our tabular perspective, this amounts to adding peak columns to the feature table while retaining the same reconstruction and contrastive objectives. As shown in **Appendix Figure 11**, Tabula performs on par with or surpasses scGPT, Geneformer and scBERT in both multi‑omics and multi‑batch integration.
>
> **Question 5**: While federated training mitigates data sharing, does the model include any formal privacy guarantees (e.g., differential privacy)? Or plans to support that?
>
> **Answer**:  We thank the reviewer for this important question. While our current implementation does not yet include formal privacy guarantees such as differential privacy (DP), we plan to incorporate such mechanisms in the next version of our framework. Tabula is fully compatible with client-side DP techniques (e.g., DP-SGD or noise-injected updates), and we view this as a promising direction—particularly in clinical or regulatory settings where stronger formal guarantees are essential.
> Notably, Tabula is, to our knowledge, the first framework to propose a federated foundation model for single-cell omics. We have observed increasing privacy concerns even in publicly available single-cell datasets [2]—a challenge likely to intensify as data scale and sensitivity grow. Existing foundation models largely overlook this issue. Tabula directly addresses this urgent need by enabling collaborative model training without data sharing. We hope this work highlights the importance of federated foundation models in single-cell research and encourages broader community participation.
>
> **Weakness 1**: While TABULA excels in single-cell applications, its applicability to other tabular biological data (e.g., clinical, bulk RNA-seq) is not explored or discussed.
>
> **Answer**: We appreciate the reviewer’s thoughtful comment. Yes, Tabula can be directly applied to other tabular biological data, such as bulk RNA-seq and multi-omics datasets. Please refer to Question 4’s answer for multi-omics data discussion. We will include additional discussion on the applicability of Tabula to other types of biological data in the revised manuscript.
>
> **Weakness 2**: The dual loss setup and federated training infrastructure introduce engineering and computational overhead; practical deployment in non-institutional settings (e.g., labs without compute clusters) could be challenging.
>
> **Answer**: We thank the reviewer for this excellent question and fully agree with the concern. Our goal is not only to propose a method but also to build a practical platform that benefits the broader research community. To support wider adoption, we will release Chiron, a distributed foundation model training platform, by the end of October. On Chiron, any user can launch a training job, and clients with private data can participate without sharing raw data. Importantly, no complex engineering setup is needed—clients simply upload their data to a local server (e.g., HPC) and run a single command to connect. Once connected, they can join any training job on the platform. Chiron enables scalable, privacy-preserving collaboration with minimal setup, and we’re extremely excited about Chiron release this year.
>
> **Weakness 3**:  Although mentioned in the appendix, the main paper could benefit from more visibility on the necessity of both reconstruction and contrastive losses.
>
> **Answer**: We thank the reviewer for this helpful suggestion. We totally agree with this. Accordingly, we will move and expand the relevant discussion from the appendix into the main text in the revised manuscript.
>
> **Weakness 4**:  While the appendix addresses limitations, more discussion in the main text would strengthen the paper’s transparency.
>
> **Answer**: We thank the reviewer for this valuable feedback. We agree with this. In the revised version, we will move and expand the relevant content from the appendix to the main manuscript.
>
> [1] Wang, Haolin, et al. "Why Go Full? Elevating Federated Learning Through Partial Network Updates." NeurIPS. 2024.
>
> [2] Walker, C. R., Li, X., Chakravarthy, M., Lounsbery-Scaife, W., Choi, Y. A., Singh, R., & Gürsoy, G. (2024). Private information leakage from single-cell count matrices. Cell, 187(23), 6537-6549.

---

> > ### Author Response · Authors · 2025-08-05
> > **Friendly Follow-Up on Tabula – Awaiting Your Feedback**
> >
> > Dear Reviewer,
> >
> > Thank you so much for your valuable comments and for recognizing the **novelty**, **effectiveness**, and **presentation** of our work. We have now addressed every question and weakness you raised:
> >
> > - **Questions 1–5:** Detailed explanations of α-sensitivity, federated-training cost, order-invariance, multi-omics extension, and future differential-privacy support.
> > - **Weaknesses 1–4:** Added discussion on broader biological data, practical deployment via our upcoming **Chiron** platform, the necessity of dual losses, and clearer presentation of limitations in the main text.
> >
> > We believe these point-by-point responses clearly resolve the concerns you highlighted.
> > For additional details, please see our full replies above. If anything remains unclear, please let us know — there are only two days left in the discussion window, and we’re happy to provide further information right away.
> >
> > We greatly appreciate your time and insights.
> >
> > Best,
> > Tabula Team

---

> > ### Author Response · Authors · 2025-08-07
> > **Friendly Reminder – 24 Hours Left for Discussion**
> >
> > Dear Reviewer,
> >
> > As the discussion window closes in less than 24 hours, we wanted to gently follow up in case there are any remaining concerns or points we can clarify. We’ve had the opportunity to engage with the other reviewers and would greatly value your final thoughts as well. Please don’t hesitate to let us know — we would be happy to respond immediately.
> >
> > Thank you again for your time and thoughtful review.
> >
> > Best regards,
> > Tabula Team

---

> > ### Author Response · Authors · 2025-08-09
> > **Final Reminder — Discussion Closing in <10 Hours**
> >
> > Dear Reviewer,
> >
> > With the discussion window closing in under 10 hours, we wanted to check once more in case there are any remaining questions we can clarify. We believe our rebuttal response addresses the concerns you raised completely; if anything is still unclear, we are on standby and can respond immediately within the remaining window.
> >
> > If our responses resolve your concerns, we would greatly appreciate a brief update or an updated evaluation at your convenience.
> >
> > Thank you again for your time and thoughtful review.
> >
> > Best regards,
> > Tabula Team

---

### Decision · Program_Chairs · 2025-09-17

**Decision:**

Accept (poster)

**Comment:**

This paper presents a foundation model designed for single-cell transcriptomics with two main contributions: a self-supervised tabular learning objective and privacy-preserving pre-training for federated learning.

All the reviewers agree that the paper presents novel and practically important contributions, and gave excellent suggestions regarding improving the manuscript and additional experimental results. They raised several important concerns regarding (a) applicability to other data/tasks particularly in OOD settings, (b) specific analyses related to biological single cell data and downstream applications and (c) additional benchmarking with other methods. These concerns have been satisfactorily addressed by the authors in their rebuttal, and 3 reviewers have accordingly increased their scores to Accept (5). The final scores (confidence in parentheses) are 3 (4), 5 (3), 5 (4), 5 (4), 4 (3).

Overall, I believe this work makes important contributions to the field of single-cell foundation models. Experimental results are convincing in supporting the claims made, especially those that were done in the rebuttal. An important point raised was regarding the writing and overstating of contributions - authors should carefully incorporate this feedback in future revisions.